



# Alternative climatic steady states for the Permian-Triassic paleogeography

Charline Ragon[1], Christian Vérard[2], Jérôme Kasparian[1], and Maura Brunetti[1]

[1]Group of Applied Physics and Institute for Environmental Sciences, University of Geneva, Bd. Carl-Vogt 66, CH-1211 Geneva 4, Switzerland
[2]Section of Earth and Environmental Sciences, University of Geneva, Geneva, Switzerland

**Correspondence:** C. Ragon (charline.ragon@unige.ch) and M. Brunetti (maura.brunetti@unige.ch)

**Abstract.** Because of spatial scarcity and uncertainties in sedimentary data, initial and boundary conditions in deep-time climate simulations lack of constraints. On the other hand, climate is a nonlinear system with a multitude of feedback mechanisms, which compete and balance in a different way that depends on the initial and boundary conditions, opening the possibility, in numerical experiments, to obtain multiple steady states under the same forcing. Here, we use the MITgcm with a coupled

atmosphere-ocean-sea ice-land configuration to explore the existence of such alternative steady states around the Permian-Triassic Boundary (PTB). We construct the corresponding bifurcation diagram accounting for processes on a timescale of thousands of years, in order to identify the stability range of the steady states and tipping points in regard to atmospheric $CO_2$ content. We find three alternative steady states with a difference in global mean surface air temperature of around 10 °C. We also investigate how these climatic steady states are modified when feedbacks acting over comparable or longer time scales

are included, in particular vegetation dynamics and air-sea carbon exchange. Our findings for multistability provide a useful framework for explaining climatic variations observed in Early Triassic geological records, and some discrepancies between numerical simulations in the literature and geological data at the PTB and its aftermaths.

## 1 Introduction

Earth's climate is a complex system that results from the balance between a spatially inhomogeneous distribution of energy

received from the Sun and dissipative processes occurring at various temporal and spatial scales (Trenberth and Stepaniak, 2004; Ghil and Lucarini, 2020). A multitude of feedback mechanisms takes place in the system, some reducing (e.g., Stefan-Boltzmann radiative feedback) and others amplifying (e.g., ice-albedo feedback) the effect of an initial change in the average temperature. Depending on how these feedbacks compensate or interact between each other, the climate system can reach different steady states (or *attractors*) under the same forcing, a phenomenon called 'multistability' (Strogatz, 1994).

Global means of state variables, like surface air temperature, change in a quasi-linear manner in response to a forcing variation when the attractor remains stable. However, abrupt (and irreversible in case of multistability) climate changes can occur when the attractor looses its stability through different tipping mechanisms (Ashwin et al., 2012; Feudel et al., 2018). In particular, a bifurcation-induced tipping (B-tipping) happens at the stability limit of an attractor, called tipping point. In this case, a shift from one attractor to another is associated to an abrupt modification of climatic conditions under a small change



in forcing, such as variations of the astronomical cycles (Crucifix et al., 2006) or tectonic movements (Raymo and Ruddiman, 1992). Noise-induced tipping (N-tipping) occurs when the variability of the dynamics on the attractor attains a critical threshold where the height of the barrier separating two basins of attraction is exceeded because of the amplification of some particular feedback mechanism, triggered by the volcanic activity (Baum and Fu, 2022) or the biological pump (Mukhopadhyay and Bhattacharyya, 2008), for example, under a spatial shift of biome types (Schneebeli-Hermann, 2020). Also shock-induced

tipping (S-tipping) can be caused by volcanic activity, or alternatively by asteroid impacts (O'Keefe and Ahrens, 1989), or any other mechanisms inducing a shift to another attractor on a shorter time scale than N-tipping. Finally, a forcing that varies on a time scale faster than the attractor internal variability can give rise to rate-induced tipping (R-tipping), even in the absence of multistability (Ashwin et al., 2017; Hoyer-Leitzel and Nadeau, 2021; Feudel, 2023; Ritchie et al., 2023).

Present anthropogenic $CO_2$ emissions are pushing the system towards critical thresholds (Lenton et al., 2008; McKay et al.,

2022). Thus, a better understanding of these tipping mechanisms becomes essential through, for example, the study of the evolution of the Earth climate in the past (Wunderling et al., 2023). Signatures of global climatic transitions are indeed found in paleoclimate proxy records for several periods of Earth's history (Messori and Faranda, 2021; Boers et al., 2022), as during the Snowball Earth episodes in the Neoproterozoic era (Hoffman et al., 1998; Pierrehumbert, 2005; Hoffman et al., 2017; Eberhard et al., 2023), the Eocene-Oligocene transition (Hutchinson et al., 2021), the glacial–interglacial cycles (Ferreira

et al., 2018; Riechers et al., 2022), the whole Cenozoic, from 66 Ma to present (Westerhold et al., 2020; Rousseau et al., 2023), or the climatic oscillations observed at the Smithian-Spathian boundary in the Early Triassic (Widmann et al., 2020), just after the Permian-Triassic mass extinction ($\sim 252$ Ma), the most severe of the Phanerozoic (Raup, 1979; MacLeod, 2014; Stanley, 2016). However, our knowledge of deep-time climates is subject to large uncertainties and their numerical modelling needs to consider, in general, a wide range of initial and boundary conditions. In this context, the framework of multistability, where an

ensemble of initial conditions is explored to find the possible attractors under the same forcing and boundary conditions, and the following construction of the so-called bifurcation diagram (BD) where the forcing is varied (Brunetti and Ragon, 2023), seems particularly useful to find possible scenarios of tipping mechanisms.

Multistability has been observed in the entire hierarchy of climate models, from energy balance models (Budyko, 1969; Sellers, 1969; Ghil, 1976; Abbot et al., 2011), to Earth models of intermediate complexity with the present Earth topogra-

phy (Lucarini et al., 2010; Boschi et al., 2013), and general circulation models with slab (Popp et al., 2016) or dynamical ocean (Ferreira et al., 2011; Rose, 2015; Popp et al., 2016; Brunetti et al., 2019; Ragon et al., 2022; Zhu and Rose, 2023; Brunetti and Ragon, 2023) using aquaplanet or idealized continental configurations.

When performing numerical simulations, only selected spatial and temporal scales can be modelled because of the unavoidable compromise between the targeted phenomena under study and computational costs. This is the reason why only a limited

number of feedbacks can in general be dynamically included in numerical simulations, depending on the chosen spatial grid resolution and the model complexity. However, less computationally expensive numerical techniques can be used, like asynchronous or 'offline' coupling (Liu et al., 1999; Foley et al., 2000) or more detailed descriptions that are only activated in the last part of the simulations.





Here, we consider the Permian-Triassic paleogeography provided by PANALESIS (Vérard, 2019b, 2021), with the advan-
tage of providing a plate tectonic reconstruction with the full seabed bathymetry as well as land topography. We explore the
existence of multiple attractors using the MIT general circulation model (Marshall et al., 1997a, b; Adcroft et al., 2004; Mar-
shall et al., 2004) in a fully coupled configuration including atmosphere, ocean, sea ice and land, as described in Section 2. We
systematically consider an ensemble of initial conditions under fixed boundary conditions and external forcing to find multiple
attractors. Then, we explore a large range of atmospheric $CO_2$ content and construct the corresponding BD in Section 3.1.1.
Since the longest timescale considered in the model is the relaxation time of deep ocean dynamics (of the order of $10^3$ yr),
we use asynchronous coupling to account for the evolution of vegetation cover in Section 3.2, and we activate air-sea car-
bon exchanges in the last part of simulation runs, as described in Section 3.3. We draw our conclusions and discuss further
developments in Section 4.

## 2    Methods

### 2.1    Model description

Numerical simulations are performed using the MIT general circulation model (MITgcm, version c67f, Marshall et al., 1997a, b;
Adcroft et al., 2004; Marshall et al., 2004) in a coupled atmosphere-ocean-sea ice-land configuration. The atmospheric module
is based on SPEEDY (Molteni, 2003) that provides a simplified description of the radiative transfer with diagnostic clouds,
convective scheme, large scale condensation, vertical diffusion and surface fluxes. The coarse vertical resolution (5 layers,
with the top one representing stratosphere) together with assumptions on parameterization schemes allow for low computa-
tional costs. The oceanic dynamic component is like in state-of-the-art climate models, with 28 vertical levels. It accounts for
tracer diffusion (Redi, 1982) and advection of geostrophic eddies (Gent and Mcwilliams, 1990; Gent et al., 1995) through the
Gent and McWilliams scheme as well as vertical mixing (Large et al., 1994). Sea ice is described by a purely thermodynamical
three-layer module (Winton, 2000), and land by a two-layer module (Campin et al., 2019). We use a cubed-sphere grid with
$32 \times 32$ points per face (CS32), which corresponds to $2.8°$ horizontal spatial resolution. Cloud albedo is varying as a func-
tion of latitude (Kucharski et al., 2013; Ragon et al., 2022) and frictional heating is re-injected into the system to guarantee
the approximate closure of the energy budget at Top-of-Atmosphere (TOA, Brunetti and Vérard, 2018; Brunetti et al., 2019;
Ragon et al., 2022; Zhu and Rose, 2023). Earth's rotation period is set to 22.2 h (Arbab, 2009) to account for Permian-Triassic
conditions. The solar constant is estimated in Foster et al. (2017) to be 1336 W m$^{-2}$.

Our simulations rely on the paleogeography produced by PANALESIS (Vérard, 2019a, b, 2021), a global plate tectonic
model with maps being created every 10 Ma from 888 Ma (Tonian) to present. The global reconstruction of the Permian-
Triassic paleogeography features a large continental mass, Pangea, extending from south to north polar regions, surrounded by
a wide oceanic realm, Panthalassa. Two more oceans are present, Tethys at equatorial latitudes and the Selwyn Sea in the north
polar region. Since narrow seaways result in unrealistic ice accumulation and numerical instabilities of the climate model,
the original PANALESIS map was adapted to the model horizontal resolution, in particular, by enlarging seaways or closing
the smallest ones, as well as lakes. The resulting topography used in the simulations is displayed in Fig. 1. Land and oceanic



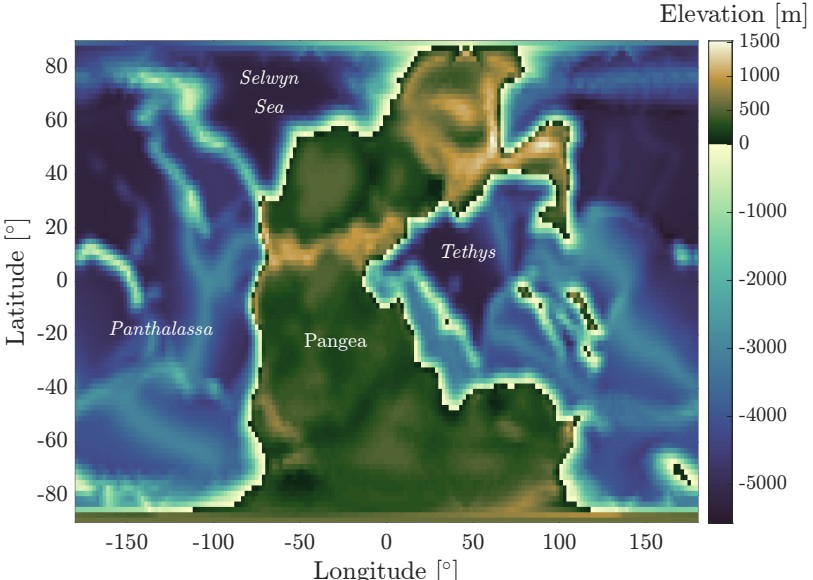

**Figure 1.** Paleogeography of the Permian-Triassic reconstruction as used in the simulations.

packages are linked via a runoff map, which is obtained by defining drainage basins and the corresponding main rivers, as represented in Fig. A1, so that each land point is associated to an ocean point that corresponds to the revelant river mouth. The initial zonal distribution of the vegetation cover is derived from Figs. 6.D-E-F of Rees et al. (2002). The corresponding values of bare soil albedos and vegetation fractions, shown in Fig. 2, are used as boundary conditions in the climate simulations.

### 2.1.1  Search for attractors and construction of BD

Estimates of the atmospheric $CO_2$ content at the Permian-Triassic boundary and the subsequent climate oscillations during the Early Triassic are affected by huge uncertainties (Retallack, 2001; Joachimski et al., 2022; Shen et al., 2022), with reported values ranging from nearly 0 ppm in the late Permian to more than 1700 ppm in the Early Triassic, according to the range of 95% confidence interval from Fig.1 of Foster et al. (2017). Our modelling strategy is thus the following: first, we set the atmospheric $CO_2$ to 320 ppm (*i. e.*, the default value in MITgcm, and we set the MITgcm parameter `aim_select_pCO2 = 1` to exclude feedback with the ocean). Using the method described in Brunetti et al. (2019), we perform dozens of simulations by varying the initial conditions[1], and let the system relax toward a climatic attractor. Second, starting from the attractors obtained at the previous step, we construct the stable branches of the BD by exploring a large range of $pCO_2$, using Method II from Brunetti and Ragon (2023). More precisely, we slightly increase or decrease the forcing by $\Delta pCO_2$ = 2–4 ppm at regular

---

[1]Either by 1) using different mean values of oceanic temperature; 2) transiently varying some parameters of internal processes, such as the relative humidity threshold for low cloud formation or atmospheric $CO_2$ content, in order to perturb the radiative budget at TOA and obtain a different climate trajectory. Afterwards, the parameters are restored to their original values.



**Figure 2.** Initial vegetation cover and albedo (top) as well as vegetation fraction (bottom) used in the climate simulations with fixed vegetation cover.





temporal intervals of $\Delta t = 100$ yr. This allows us to construct the complete BD and identify, in particular, the position of B-tipping.

## 2.2 Evolution of the vegetation cover

After having identified the attractors of the climate system, we use asynchronous coupling between MITgcm and BIOME4
models to estimate the effect of the evolution of vegetation distribution on each attractor.

BIOME4 is a coupled carbon and water-flux model (Haxeltine and Prentice, 1996; Kaplan, 2001; Kaplan et al., 2003) that is driven by long-term averages of monthly mean temperature, sunshine and precipitation to identify 27 different biomes for a given value of atmospheric $CO_2$ content. In addition, BIOME4 requires information related to soil texture and soil depth, for which we use global average of typical present-day values provided by BIOME4 itself, namely water holding capacity
(110.1 mm m$^{-1}$ for the first soil layer and 137.6 mm m$^{-1}$ for the second one) and percolation rate (5.2 mm hr$^{-1}$ for both layers).

From the attractors found at pCO$_2$ = 320 ppm, we extract monthly means of precipitation, sunshine (defined as the complement of cloud fraction, an output of MITgcm) and surface air temperature (SAT). Theses average values are given as inputs to BIOME4. After running BIOME4, the resulting biomes are converted back into albedo and fraction of soil covered by vegeta-
tion (see Table 1 for correspondences). We then restart the MITgcm from the same initial state on the attractor with these new boundary conditions, and run it further over 600 yr which corresponds to an approximate equilibrium. Monthly means are then calculated over the last 300 yr to exclude the first part of the simulation where there is an adjustment to the new vegetation cover, and used as boundary conditions to run again BIOME4. This procedure is repeated until convergence between the two models is observed, defined as 1) global SAT does not evolve anymore between two iterations, and 2) the land surface fraction
where albedo varies between two iterations is lower than 10%.

## 2.3 Air-sea carbon flux

Exchanges of carbon between the ocean and the atmosphere are particularly strong in regions of deep-water formation, upwelling or boundary currents. Since these processes regulate the climate over time scales of the order of $10^3$-$10^4$ yr (Zhu and Rose, 2023), they can affect the BD based on deep-water dynamics. Including dynamical carbon exchanges between ocean
and atmosphere (GCHEM, DIC and PTRACERS modules in MITgcm, Dutkiewicz et al., 2005; Follows et al., 2006) makes the CPU time to double compared to the configuration without, thus preventing the use of this option from the start for the construction of BDs. The less consuming option we choose is to activate the dynamical carbon cycle after the convergence of the simulations without carbon exchanges, at some particular position on each stable branch, namely, at pCO$_2$ = 320 ppm and near the edges of the stable branches. The new configuration now includes the evolution of five passive tracers: dissolved
inorganic carbon (DIC), dissolved inorganic phosphorus, alkalinity, phosphate and oxygen. For this study, we use the version c67j of MITgcm, where a new implementation of the pH solver allows for the modelling of more extreme conditions of ocean temperature than the present-day ones (Munhoven, 2013).

The numerical procedure is divided in two steps:





1.  we use the reference profiles of the five passive tracers for the present-day Earth provided by MITgcm as initial conditions

at 320 ppm. We then compute equilibrium distribution of tracers by keeping the atmospheric carbon content to 320 ppm (this is done by setting the model parameters `aim_select_pCO2` = 1 and `dic_int1` = 1). At the edges of the stable branches, equilibrium concentrations obtained for $pCO_2$ = 320 ppm are used as initial conditions to reduce spin-up time. We consider that an equlibrium is reached when the annual CO2 flux at the ocean surface is lower than 0.1 ppm $yr^{-1}$.

2.  The obtained equilibrium distributions are then used to restart the model with a fully active carbon cycle. In this case,

by setting the model parameters `aim_select_pCO2` = 3 and `dic_int1` = 3, the DIC module calculates the air-sea $CO_2$ flux at the sea surface and provides the evolution of oceanic as well as atmospheric $pCO_2$, and the global carbon content[2].

## 3    Results

### 3.1    Three attractors under a fixed atmospheric $CO_2$ content of 320 ppm

Under the same forcing represented by an atmospheric $pCO_2$ of 320 ppm, the system relaxes over a time scale of the order of several thousands of years toward either of three climatic attractors. We show here the diagnostics calculated over the last 30 yr of simulation for each attractor. Figure 3 shows sea-surface temperature and sea-ice thickness, while Fig. 4 displays the surface air temperature (SAT). Global mean SAT ranges from 30.9 °C for the *hot state* where no ice is present, to 21.56 °C for the intermediate *warm state* where a small ice cap reaches $\sim 80°$ N, and to 17.20 °C for the *cold state* with a perennial ice

cap in the northern polar region down to $\sim 62°$ N. A seasonal, small and thin sea ice layer forms in the southern polar region in both warm and cold states (see Table 2, and note that the smallest model grid-cell area is $\sim 0.03 \cdot 10^6$ $km^2$, thus ice extent smaller than this value is not relevant and is set to zero).

Table 3 shows global averages of selected key state variables and conservation diagnostics. The three attractors have a closed surface energy balance over ocean (*i. e.*, lower than 0.1 W $m^{-2}$ in absolute value), which is the dominant component as oceans

cover 69% of the Earth surface at PTB. The top-of-atmosphere (TOA) energy budget is also well closed, ranging from $-0.1$ to $-0.4$ W $m^{-2}$, and the water-mass budget is null in the three simulations. This is confirmed by the fact that the global ocean temperature and the global mean salinity show negligible drifts over the last 100 yr of simulation, namely, less than $1.5 \cdot 10^{-2}$ °C and $8 \cdot 10^{-3}$ psu (in absolute value), respectively.

As reported in an aquaplanet configuration (Ragon et al., 2022), the presence of sea ice is associated with a stronger heat

transport, and by the increase of the meridional surface air temperature gradient, defined as the temperature difference between polar (30° to 90°) and equatorial (-30° to 30°) regions. This is indeed what is observed in the northern hemisphere, where the meridional temperature gradient is larger in cold (28.5 °C) than in warm (22.3 °C) and hot (17.2 °C) states. Thus, the

---

[2]Note that, by setting initial tracers concentrations to values different from the equilibrium distributions obtained at step 1, for example directly using the reference MITgcm profiles, a spurious amount of carbon would enter the global carbon reservoir and create a perturbation that could push the system into another attractor.



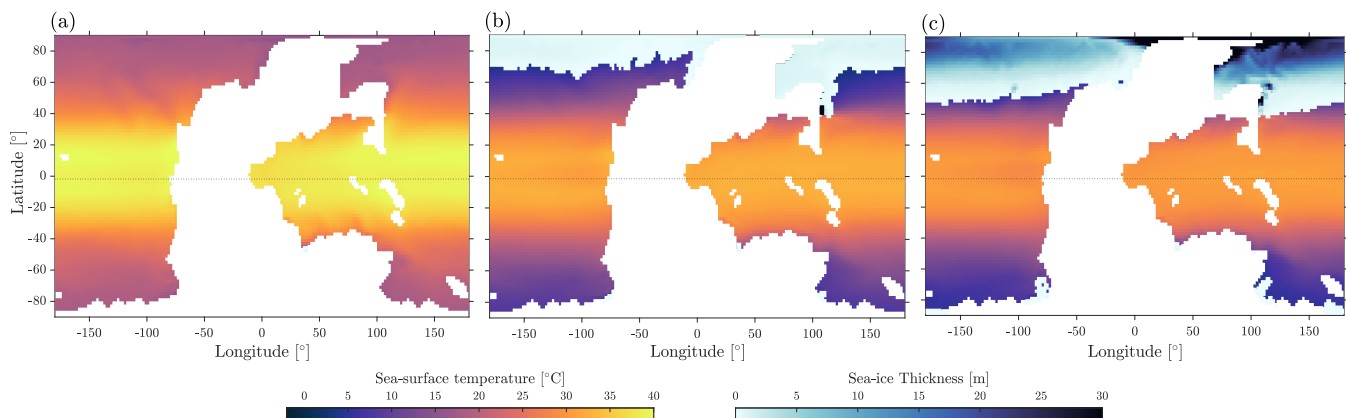

**Figure 3.** Sea-surface temperature and sea-ice thickness for (a) hot, (b) warm, and (c) cold states. White area corresponds to land.

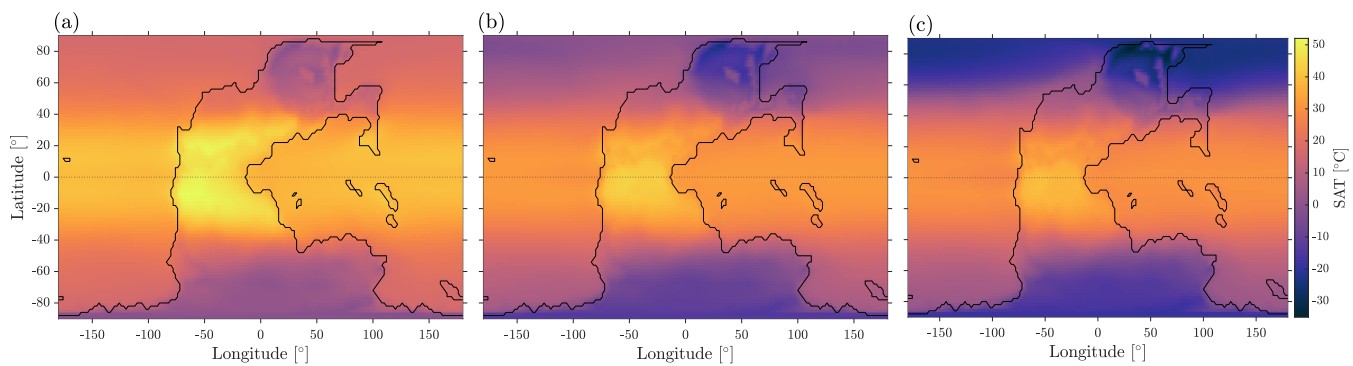

**Figure 4.** Near-surface air temperature for (a) hot, (b) warm, and (c) cold states.

atmospheric heat transport is more intense in the cold state in the northern hemisphere, as seen in Fig. 5a, while in the southern hemisphere, where ice caps are absent or negligible, the meridional temperature gradient is nearly equal in all states, implying a small variation in heat transport efficiency. In this case, other contributions to the total heat transport need to be accounted for to differentiate the states, as the latent heat and the associated water-mass transport (see Fig. 5b) which turns out to be more important in the hot state, where the whole hydrological cycle has larger intensity (*i. e.*, more evaporation and precipitation, as listed in Table 3). We also observe an asymmetry between the northern and southern hemispheres in the atmospheric overturning circulation, as shown in Fig. 6. In the northern hemisphere, the Hadley cell moves equatorward and becomes stronger in the cold state (see Fig. 6a-c), leading to an increase of $\sim 3 \cdot 10^8$ kg s$^{-1}$ in the equatorward moisture transport peak (Fig. 5b). Note that, when oceanic circulation is vigorous like in the hot state (Fig. 6d), the atmospheric circulation is weak (Fig. 6a) compared to other attractors, while a weak oceanic circulation leads to a strong atmospheric one, as seen in the cold state (Fig. 6c and 6f). This is the well-known phenomenon denoted as Bjerknes compensation (Bjerknes, 1964).



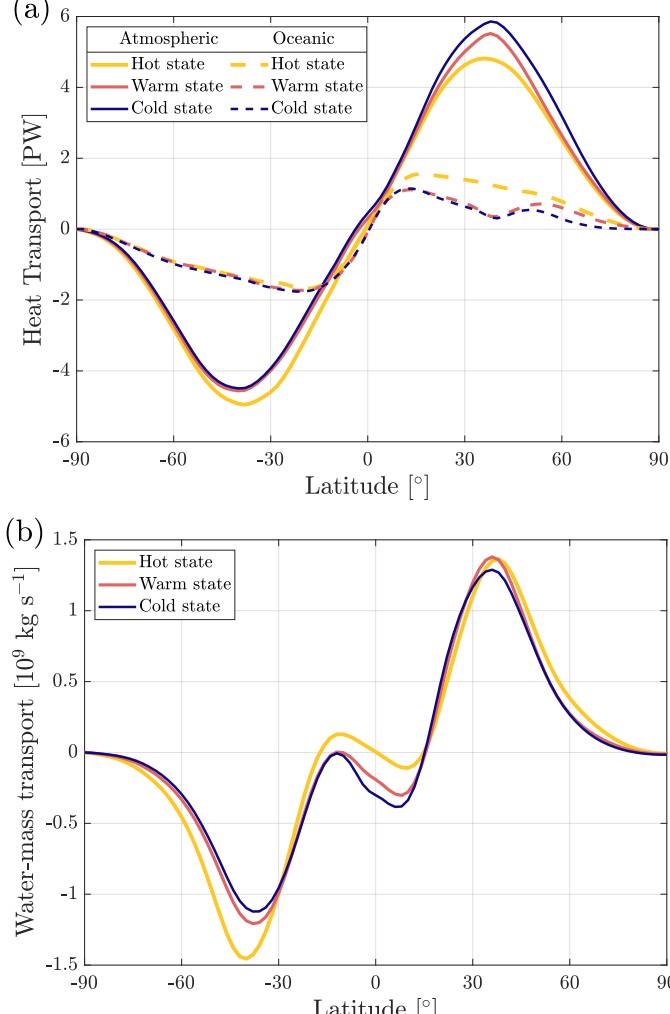

**Figure 5.** Climatological annual mean of northward meridional transport of (a) heat in the atmosphere (solid lines) and ocean (dashed lines), and of (b) water mass in the atmosphere, for the three attractors.

Evaporation is maximal on oceanic tropical regions, while it is nearly zero over polar regions in the presence of sea ice, as seen in Fig. 7. Largest rate of precipitation (Fig. 8) is observed in equatorial regions, as in present-day climate. In the hot state, the amount of precipitation falling on land in the north polar region remains important. Dry continental areas, with low precipitation and evaporation, are present along the tropics in the three attractors.

Salinity (see surface distribution is shown in Fig. 9) is correlated to the difference between evaporation, precipitation and runoff. In hot state, where sea ice is absent, the salinity distribution is symmetrical with respect to the Equator as well as the sea surface temperature (Fig. 3), giving rise to a symmetrical configuration of the deep-water convection regions at the poles and of the oceanic overturning cells, with strong upwelling at the Equator (Fig. 6d). This symmetry disappears in cold and warm





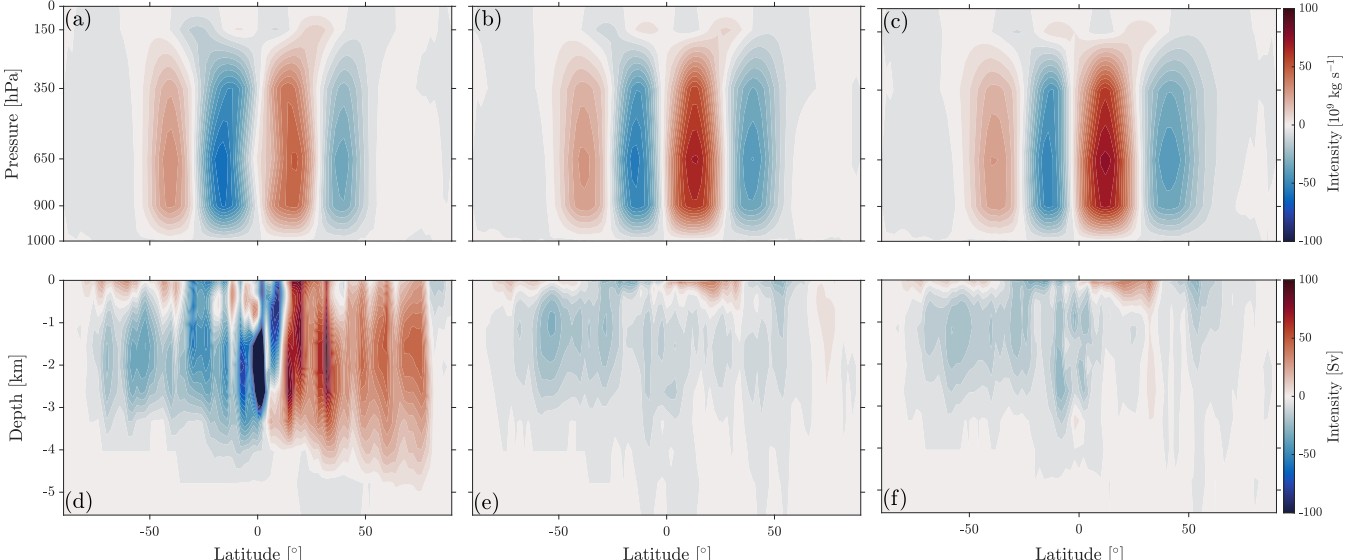

**Figure 6.** Global overturning circulation in the atmosphere (a-c) and in the ocean (d-f) for (a,d) hot (b,e) warm and (c,f) cold states. Color indicates streamfunction strength. Units are $10^9$ kg s$^{-1}$ for atmosphere and Sverdrups (1 Sv $= 1 \cdot 10^6$ m$^3$ s$^{-1}$) for ocean. Positive and red (*resp.* negative and blue) corresponds to clockwise (*resp.* anticlockwise) circulation. Each color gradation corresponds to the transport of 5 Sv (ocean) and 5 m$^3$ s$^{-1}$ (atmosphere).

states (Fig. 6e and 6f), where the absence of the northern overturning cell is associated with a drop of salinity and the presence of sea ice. In this case, the anti-clockwise cell extends in the two hemispheres, with a largely reduced intensity compared to the hot state. It is interesting to note that Hülse et al. (2021) show similar patterns for the overturning cells at PTB, for different

values of atmospheric $CO_2$ content: the clockwise overturning cell becomes intense as the $CO_2$ increases, in agreement with the behavior we find in the hot state. Contrary to the thermohaline circulation, the overall structure of surface currents (Fig. 10) is similar in the three attractors, although the subtropical gyres are more symmetrical and intense in the two colder states. In terms of oceanic circulation, warm and cold states display a similar dynamics.

### 3.1.1    Bifurcation diagram for varying atmospheric content of $CO_2$

The three attractors described until now have been characterized for p$CO_2$ = 320 ppm. By varying the atmospheric $CO_2$ content and thus modifying the radiative forcing, we find the stability regions of each attractor and the associated tipping points through the construction of the bifurcation diagram (BD) following Method II in Brunetti and Ragon (2023) (see Section 2.1.1 and Fig. B1). The resulting stable branches are shown in Fig. 11[3]. Tipping points are located at the endpoints of the stable branches, except for the right end of the stable range for the hot state, which could not be reached. In fact, even by increasing

---

[3]Climatic attractors are complex dynamical objects living in a high-dimensional manifold (Brunetti et al., 2019; Falasca and Bracco, 2022). The projection of their invariant (or natural) measure (Eckmann and Ruelle, 1985) on a given state variable is arbitrary (Faranda et al., 2019; Tél et al., 2020; Brunetti and Ragon, 2023). In the present study, the projection is performed, as commonly done in the literature, in terms of the global mean SAT.





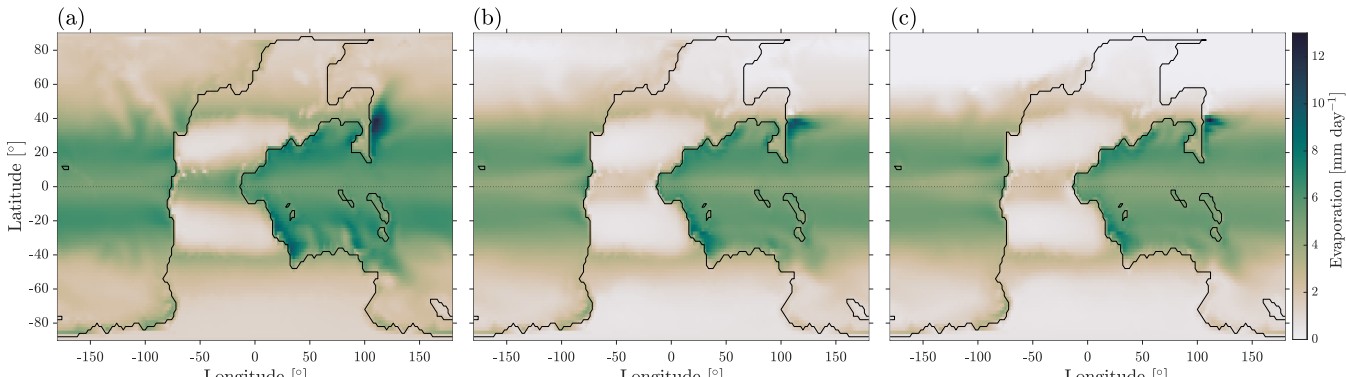

**Figure 7.** Evaporation for (a) hot, (b) warm and (c) cold states.

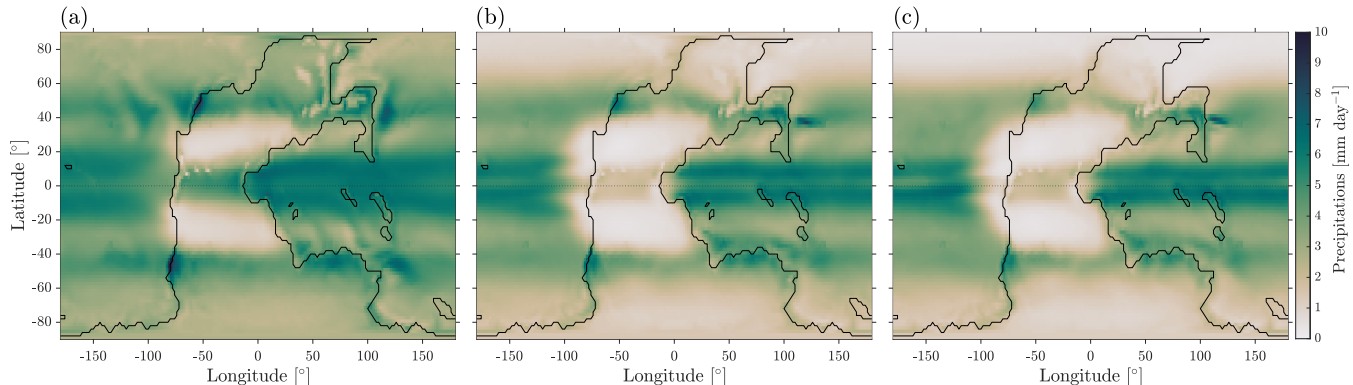

**Figure 8.** Precipitation for (a) hot, (b) warm, and (c) cold states.

the forcing up to $\sim 500$ ppm, we do not find a tipping point when SAT increases toward values larger than 38 °C. Rather, for these values, the internal variability becomes very large, suggesting that the model becomes numerically unstable and is unable to describe such extreme conditions. The limitation is due to the high temperature rather than to the $pCO_2$ value, since previous works using the MITgcm for an aquaplanet managed to simulate twice larger $pCO_2$ values, although for lower values of SAT (Brunetti and Ragon, 2023).

The region of tristability extends between 320 and 328 ppm and corresponds to the extent of the warm state branch. Both hot and cold branches extend beyond the warm one. Thus, through a B-tipping, it is possible to tip from warm to cold or hot states, but impossible to reach the warm state from either cold or hot states, which excludes the possibility of an hysteresis loop involving the warm state. The only way to attain the warm state would be to initially stay into that state, or to reach it through another kind of tipping mechanism. However, if we assume that the size of the warm state basin of attraction is proportional to

the branch length, catching the warm attractor would require very specific initial conditions. This suggests that the warm state may only have a low impact on the general dynamics of the climate system.



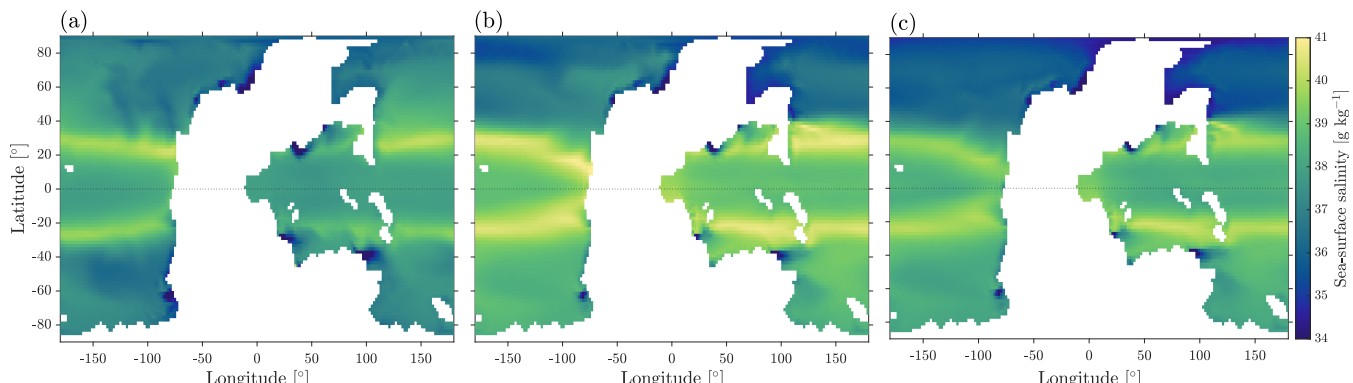

**Figure 9.** Sea-surface salinity of (a) hot, (b) warm, and (c) cold states. White area corresponds to land.

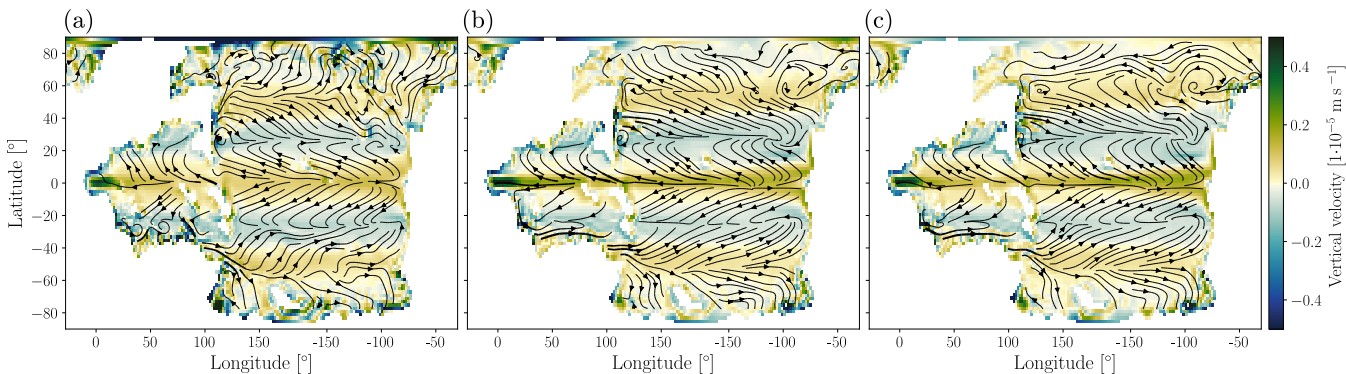

**Figure 10.** Surface oceanic circulation in (a) hot, (b) warm and (c) cold states. Black arrows correspond to mean horizontal circulation over the first 50 m. The line thickness varies with the horizontal velocity. The colorscale represents the vertical velocity: positive (green) are upwelling areas, negative (blue) are downwelling.

A larger region of bistability from 304 to 332 ppm allows for the presence of an hysterisis loop between the hot and cold states. The temperature gap between these two attractors is of the order of 10 °C. Thus, our numerical results suggest that tipping mechanisms can induce a shift from one attractor to another with climatic variations of this order of magnitude. It is

interesting to note that there is a similar temperature gap of 10 °C between previously published model outputs and geological records for the Permian-Triassic period, as shown in Lunt et al. (2023, slide 14), Scotese et al. (2021).

The BD of Fig.11 gives essential information on the climate system, over the dynamical timescales considered in the numerical model. The longest one, corresponding to deep-ocean dynamics, is of the order of $10^3$ yr. Additional feedback mechanisms developing over similar or longer timescales need to be investigated using different numerical techniques, as discussed in the

following subsections.





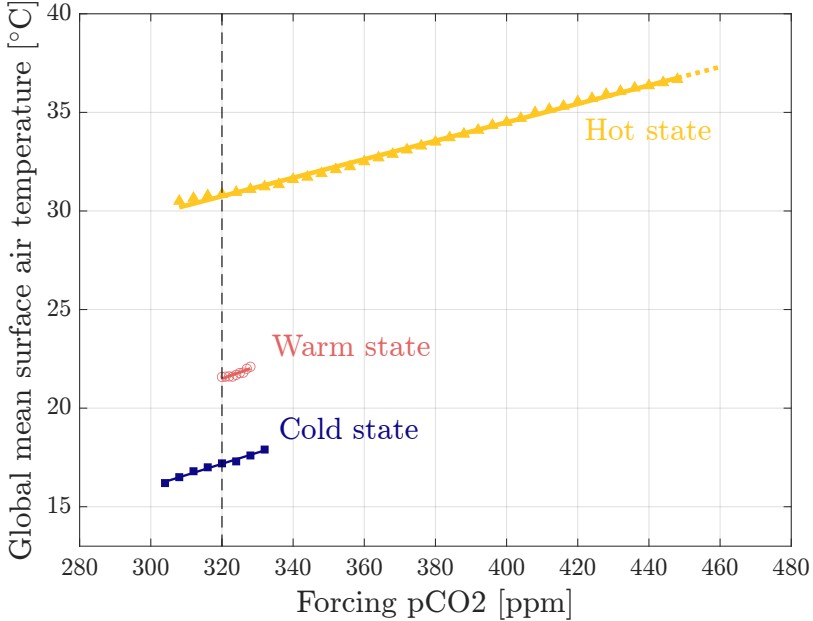

**Figure 11.** Bifurcation diagram as a function of $pCO_2$ forcing. Markers correspond to averages over 100 yr for a given forcing value, error bar are within the marker size. Solid lines are the linear fit of each branch, slopes $s$ being similar: $s_{hot} = 0.0469$ (5) °C ppm$^{-1}$, $s_{warm} = 0.06$ (1) °C ppm$^{-1}$ and $s_{cold} = 0.057$ (3) °C ppm$^{-1}$. Vertical dashed black line indicates the reference value $pCO_2 = 320$ ppm discussed in Section 3.1.1.

## 3.2 Vegetation cover on land

Vegetation distribution provides an important feedback mechanism to the climate system, since it affects the albedo as well as evapotranspiration over land surfaces, and thus the radiative budget. Moreover, the distribution and amount of terrestrial biomass play a crucial role in the global carbon cycle. Here, we evaluate the long-term climate adjustment (up to $10^4$ yr, McGuffie and Henderson-Sellers, 2005, Fig. 2.12) due to the vegetation cover associated to each attractor set at a $pCO_2$ forcing of 320 ppm. To satisfy the convergence criteria reported in Section 2.2, five iterations were needed for the hot state, four for the cold state and three for the warm states. SAT and percentages of land surface with albedo variations are listed in Table 4 for each iteration. Taking the evolution of the vegetation cover into account does not affect the number of attractors but induces a shift in their average SAT, which increases by 1.5 °C in hot, 0.2 °C in warm and 0.8 °C in cold state. Figure 12 shows the resulting biome maps for the three attractors. For sake of clarity, biomes are grouped into major biomes (see classification in Table 3 of Harrison and Prentice, 2003). Note that biomes 16 (temperate broadleaved savanna) and 18 (boreal parkland) do not appear in our simulations.





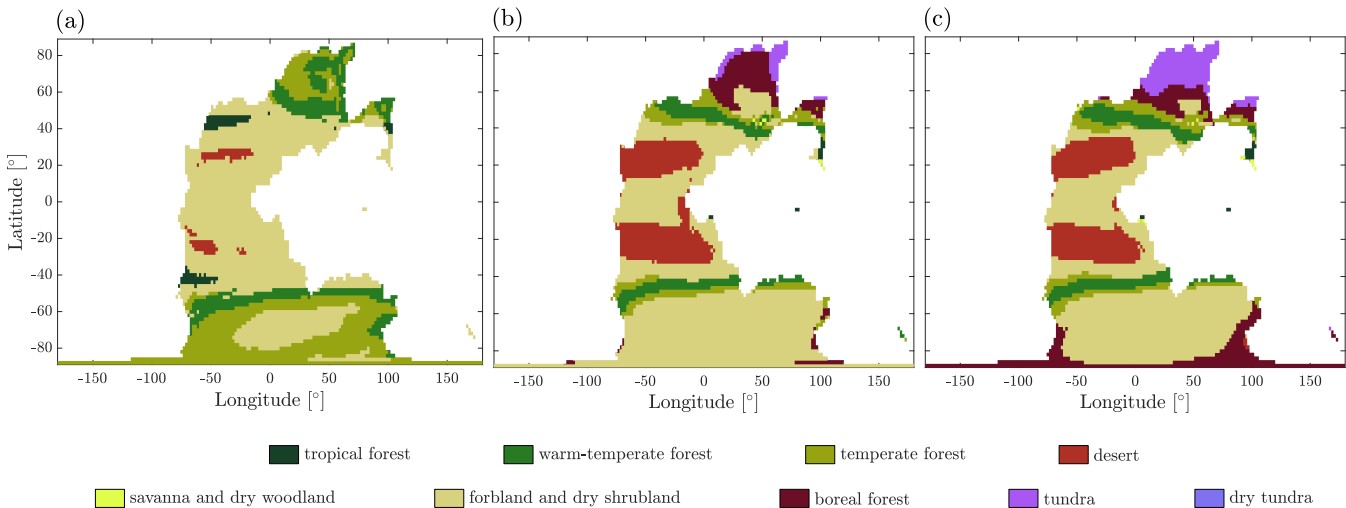

**Figure 12.** Vegetation cover represented through major biomes (see legend, Harrison and Prentice, 2003) for (a) hot, (b) warm, and (c) cold states, with atmospheric $CO_2$ content of 320 ppm. White area corresponds to ocean.

In the cold state, desert areas are present at the tropics, surrounded by forbland[4]. Moving poleward, the vegetation successively evolves to temperate/warm-temperate forests, to forbland again and to boreal forests. The forbland area is much larger
in the southern hemisphere, covering almost all longitudes between $60°$S and $80°$S. Tundra is present in the north polar region, developing at the same latitudes as the sea ice extent. This suggests that ice sheets could develop instead of tundra if included into the simulation setup, thus albedo would be larger and the resulting climate even colder. The warm state displays a similar trend, with the main differences occurring in polar regions: in the north, boreal forests dominate over tundra, while in the south, forests are mainly replaced by forbland. Finally, in the hot state, desertic areas nearly disappear, and forbland dominates
in both tropical and subtropical regions. In fact, there is more precipitation in hot state (see Fig. 8 and Table 3) preventing the development of desert despite high air temperatures. Polar regions are covered by successive slices of warm-temperate and temperate forests, as well as forbland in the south.

In order to estimate the amount of terrestrial biomass in each attractor, we use the correspondence between ecosystem type and mean biomass density as reported in Table 1 of Houghton et al. (2009). The 28 biomes in BIOME4 have been associated
with the ecosystem type as shown in Appendix C. We find that the mean biomass density in hot state is of the order of 145 Mg ha$^{-1}$ (corresponding to a total of $1.9 \cdot 10^{17}$ mol of carbon globally), which is larger than 92 Mg ha$^{-1}$ ($1.2 \cdot 10^{17}$ mol) in cold state. While these values lie a factor of 4 above present-day values (Bar-On et al., 2018) and may be overestimated, they clearly demonstrate the trend that the organic carbon stored in the hot state vegetation cover is larger than in colder climates.

---

[4]The present-day 'grassland' is replaced by the herbaceous non-graminoid 'forbland' because graminoids appeared during the Cenozoic (Gradstein and Kerp, 2012).



### 3.3 Air-sea carbon exchange

The carbon cycle describes how carbon is transferred between the atmosphere, hydrosphere, biosphere, cryosphere and lithosphere over different timescales (Dolman, 2019, Chapter 9). In our simulation setup, we can include how the carbon flows between atmosphere and ocean as described in Section 2.3, but not among the other reservoirs. Once equilibrium is reached, the carbon cycle acts to maintain a net zero flux at the interface between atmosphere and ocean, and thus regulates the climate over $10^3$-$10^4$ yr. Here, we evaluate the implications of these carbon exchanges on the structure of the BD.

By including carbon exchanges between atmosphere and ocean in simulations at 320 ppm and at the edges of the stability branches, we obtain the values listed in Table 5. The advantage of including air-sea carbon exchanges is that DIC module estimates the total carbon stored in the system and the corresponding equilibrium temperature. We can see that including a feedback between the two reservoirs both the atmospheric $CO_2$ content and the equilibrium SAT are slightly changed. However, the overall pattern of the BD is not affected, as shown in Appendix D.

Even if we have included the exchange of carbon between atmosphere and ocean, it is important to remember that our simulation setup does not take into account carbon fluxes between lithosphere, biosphere and cryosphere, such as weathering, biomass in terrestrial plants or stored in the permafrost. Thus, the 'total' carbon that we have estimated is affected by these simplifications. Since the precipitation (see Fig. 8) and mean biomass quantity (see Section 3.2) are much larger in hot than in cold state, neglecting weathering and biomass fluxes has a larger impact on hot than cold climates. In particular, differences in biomass carbon, which we estimate in the range of $10^{17}$ mol in Section 3.2, lies in the same range as the difference between the amounts of 'total' carbon content in the cold state ($3.0 \cdot 10^{18}$ mol) and the hot state ($2.8 \cdot 10^{18}$ mol). Since vegetation and air-sea carbon exchanges act on the same time scale, the biomass could significantly contribute to this difference.

Considering the atmosphere-ocean $CO_2$ coupling further reduces the range of stability of the warm state, which almost collapses to a single point around 320 ppm. Thus, this climatic state occupies a very small dynamical phase space region that can only be attained from specific initial conditions.

### 4 Conclusions

The climate is a nonlinear system that can display multistability, reflecting the fact that there is no a unique way to redistribute energy when several feedbacks are active on the same time scale. Under a variation of forcing such as atmospheric $CO_2$ content, the climate can shift from one attractor to another. This transition is explained by a modification of the balance between feedback mechanisms (Lucarini et al., 2014). Bifurcation diagrams (BD) are an important tool to clarify the phase space of the system that can include several attractors, in particular for identifying stability and multistability regions, position of tipping points, as well as amplitude of forcing and of internal variability needed to allow for a transition between attractors.

We performed numerical simulations using the MIT general circulation model with a coupled atmosphere-ocean-sea ice-land configuration and the Permian-Triassic paleogeography provided by PANALESIS. We found three attractors with global mean SAT ranging from 17 to 31 °C at $CO_2$ = 320 ppm, named cold, warm and hot state. We constructed the BD, for pCO$_2$ ranging between 300 and 460 ppm (Fig. 11). Over this range, we identified bistable and tristable regions.



In the configuration used, feedback mechanisms occurring on time scales comparable or longer than the relaxation time of the deep-ocean circulation are excluded and need to be accounted for using different numerical techniques. We restarted simulations at selected positions in the BD (i.e., 320 ppm and edges of the stable branches) by including relevant long-term feedbacks, namely the evolution of the vegetation distribution (through asynchronous coupling between MITgcm and BIOME4) and the exchange of carbon between atmosphere and ocean (by the activation of additional packages in MITgcm). These methodologies allowed us to include long-term adjustments and to correct the position of attractors' stable branches that turn out to be only slightly modified. Analogous techniques could be applied to account for any other relevant long-term mechanisms, like the evolution of ice-sheets, which is presently not implemented in our setup.

While absolute values are of little relevance because they are highly sensitive to the numerical model, the general trend shows that multistability can exist in a general circulation model with a realistic paleogeography. By displaying a temperature gap of the order of 10 °C between hot and cold states, the BD corresponding to the Permian-Triassic paleogeography opens the possibility to explain some climatic variations observed in geological records of the Early Triassic by the hysteresis loop existing between these two attractors (B-tipping), by increased internal variability due to the biological pump or volcanism (N or S-tipping), or by a forcing mechanism which varies in time with a critical rate (R-tipping). A temperature difference of 10 °C between climatic states also provides a possible explanation for the difference between surface temperatures reconstructed by geological records (Scotese et al., 2021) and those obtained by numerical models (Valdes et al., 2021; Li et al., 2022; Lunt et al., 2023).

BDs constructed by using other numerical models with the same feedback mechanisms would be necessary to find robust characteristics, and to reduce artefacts of models and boundary conditions. The construction of BDs is a promising method for investigating the dynamics of the climate system in deep time, as well as in the present day forcing setup.

*Code and data availability.* The data that support the findings of this study were generated by the MIT general circulation model that is openly available on GitHub (http://mitgcm.org/,https://github.com/MITgcm/MITgcm, versions c67f and c67j), and by BIOME4 (https://github.com/jedokaplan/BIOME4).



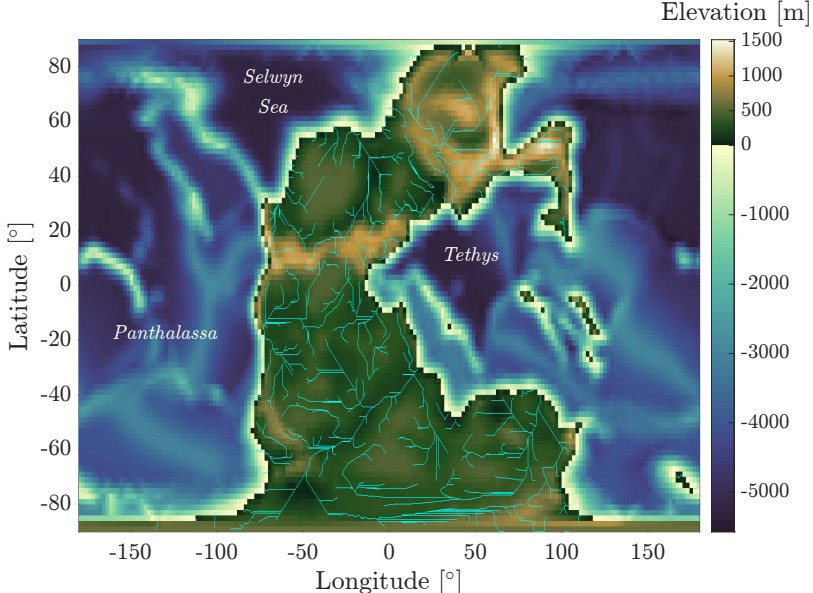

**Figure A1.** Rivers (cyan lines) map for the Permian-Triassic paleogeography reconstructed after PANALESIS.

## Appendix A: Runoff in MITgcm

Land and oceanic modules in MITgcm are linked through the runoff map, where each land point in watersheds is connected to an oceanic point, which represents the corresponding river mouth. Figure A1 shows the main rivers, in cyan, for the Permian-
Triassic paleogeography.

## Appendix B: Construction of the bifurcation diagram

We use Method II described in Brunetti and Ragon (2023) for obtaining the bifurcation diagram. In particular, once the three attractors at 320 ppm are found by scanning a large ensemble of initial conditions (see main text, Section 3.1), the atmospheric pCO$_2$ content is varied by $\Delta$pCO$_2$ = 2–4 ppm at regular temporal intervals of $\Delta$t = 100 yr starting from each attractor, in
order to construct the respective stable branches (Fig. B1). Tipping points are found when one of the following three criteria is satisfied (Brunetti and Ragon, 2023): (*i*) increased standard deviation of the mean SAT; (*ii*) surface energy unbalance; (*iii*) monotonic shift toward another attractor during the time interval $\Delta$t.





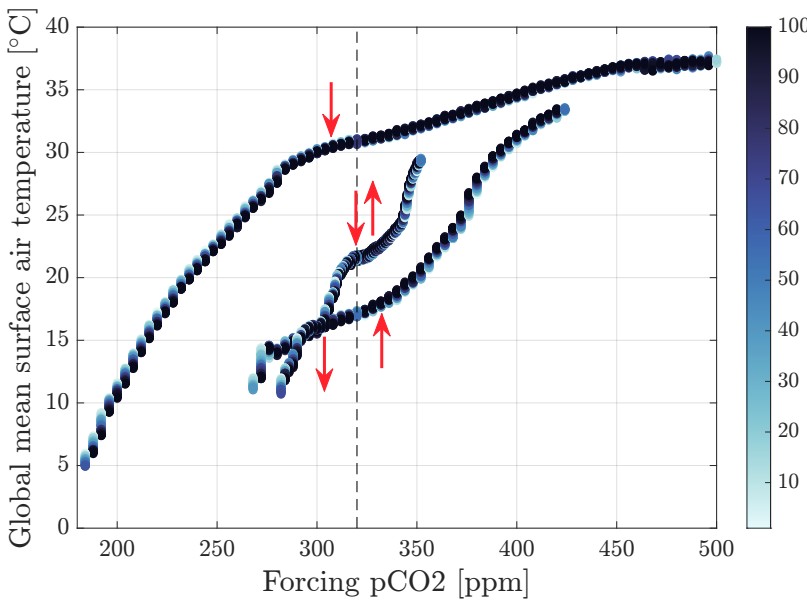

**Figure B1.** Bifurcation diagram obtained using Method II in Brunetti and Ragon (2023) by varying atmospheric $pCO_2$. Color bar refers to the time (in years) since the last change in the forcing. Red arrows corresponds to tipping point positions along the stable branches.

## Appendix C: Biomass density

Global mean biomass density is estimated for the three attractors at 320 ppm using values for typical ecosystem types reported in Houghton et al. (2009). Where several values are provided, the one from Saugier et al. (2001) is used. Table C1 shows the correspondence between the biomes used in BIOME4, ecosystem types and biomass density.

## Appendix D: Bifurcation diagram corrected by including air-sea carbon exchange

Carbon cycle is partly included by allowing exchanges of carbon between atmosphere and ocean, as described in Section 2.3. By including this feedback on simulations at 320 ppm and the edge of the branches, we can estimate the modification of the BD structure, as shown in Table 5 and Fig. D1.

*Author contributions.* M. B. planned the study, C. V. provided the paleogeographic reconstruction, C. V. and C. R. obtained the runoff map, C. R. and M. B. performed the climate simulations, C. R. produced the final numerical results and figures, all the authors were involved in the analysis and discussion of the results. C. R. and M. B. wrote the manuscript, and all the authors discussed and edited the final paper.

*Competing interests.* The authors declare that they have no conflict of interest.



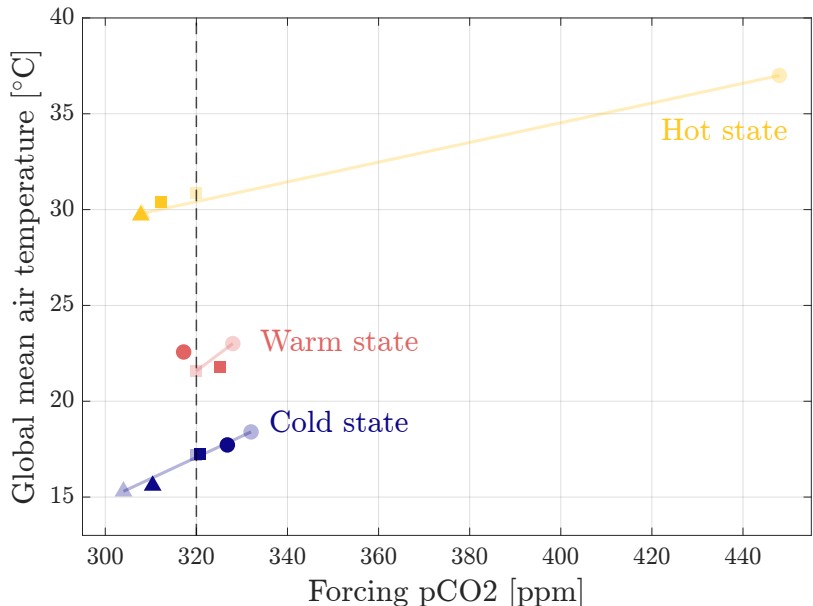

**Figure D1.** Bifurcation diagram including air-sea carbon exchanges (dark colors) compared to the original version presented in Fig. 11 (shaded colors). Squares correspond to the simulations started at 320 ppm, triangles to the ones at the left edges of the stable branch, and circles to the ones at the right edges.

*Acknowledgements.* We are grateful to Jan-Henrik Malles for running some of the simulations in the first stage of this work. We thank Stéphane Goyette and all the Sinergia project members (PaleoC4, https://www.unige.ch/paleoc4/) for very useful discussions. C. R. and M. B. thank the MITgcm-support mailing list for valuable advice on the code. The simulations were performed on the Baobab and Yggdrasil clusters at the University of Geneva. We acknowledge the financial support from the Swiss National Science Foundation (Sinergia Project No. CRSII5_180253).



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



**Table 1.** Albedo and fraction of soil covered by vegetation for each biome from BIOME4 model output. The group of biomes used as initial conditions in Fig. 2 are in italic.

| Number and Name | | Albedo [%] | Vegetation Fraction [%] |
|---|---|---|---|
| *Cool temperate* | | | |
| 1 | Tropical evergreen broadleaf forest | | |
| 2 | Tropical semi-evergreen broadleaf forest | | |
| 3 | Tropical deciduous broadleaf forest and woodland | | |
| 4 | Temperate deciduous broadleaf forest | | |
| 5 | Temperate evergreen needleleaf forest | 13 | 90 |
| 6 | Warm-temperate evergreen broadleaf and mixed forest | | |
| 7 | Cool mixed forest | | |
| 8 | Cool evergreen needleleaf forest | | |
| 9 | Cool-temperate evergreen needleleaf and mixed forest | | |
| 17 | Temperate evergreen needleleaf open woodland | | |
| *Tropical summerwet* | | | |
| 12 | Tropical savanna | 16 | 95 |
| *Oceanic influenced dry zone* | | | |
| 13 | Tropical xerophytic shrubland | | |
| 14 | Temperate xerophytic shrubland | 19 | 50 |
| 15 | Temperate sclerophyll woodland and shrubland | | |
| 20 | Temperate grassland | 23 | 90 |
| *Tundra* | | | |
| 22 | Graminoid and forb tundra | | |
| 23 | Low and high shrub tundra | | |
| 24 | Erect dwarf-shrub tundra | | |
| 25 | Prostrate dwarf-shrub tundra | 24 | 20 |
| 26 | Cushion-forb tundra | | |
| 10 | Cold evergreen needleleaf forest | | |
| 11 | Cold deciduous forest | | |
| 19 | Tropical grassland | 26 | 90 |
| *Desert* | | | |
| 21 | Desert | 31 | 1 |
| 27 | Barren | | |





**Table 2.** Maximal and minimal seasonal sea ice area $A$ [$10^6$ km$^2$] and thickness $H$ [m] for each hemisphere and associated standard deviation derived from annual variability in parentheses

| | *Northern Hemisphere* | | | |
| | A$_{min}$ | H$_{min}$ | A$_{max}$ | H$_{max}$ |
|---|---|---|---|---|
| Hot state | 0 | 0 | 0 | 0 |
| Warm state | 0.4 (2) | 2.1 (1) | 10.6 (4) | 26 (15) |
| Cold state | 27.2 (3) | 7.29 (9) | 40.2 (4) | 10.2 (2) |
| | *Southern Hemisphere* | | | |
| | A$_{min}$ | H$_{min}$ | A$_{max}$ | H$_{max}$ |
| Hot state | 0 | 0 | 0 | 0 |
| Warm state | 0 | 0 | 0.09 (6) | 0.6 (4) |
| Cold state | 0 | 0 | 0.19 (8) | 0.5 (2) |





**Table 3.** Global mean values averaged over the last 30 yr of simulation and associated standard deviation in parentheses derived from inter-annual variability, for each attractor. NH: northern hemisphere, SH: southern hemisphere.

|  | Hot state | Warm state | Cold state |
| --- | --- | --- | --- |
| *Conservation diagnostics* | | | |
| TOA imbalance [W m$^{-2}$] | −0.1 (1) | −0.4 (1) | −0.3 (2) |
| Ocean surface imbalance [W m$^{-2}$] | 0.0 (2) | −0.1 (2) | 0.07 (3) |
| Water-mass budget [$10^{-5}$ g m$^{-2}$ s$^{-1}$] | 0 (2) | 0 (2) | −0 (2) |
| Ocean temperature drift [°C century$^{-1}$] | 0.005 (1) | −0.015 (1) | 0.006 (2) |
| Salinity drift [$10^{-3}$ psu century$^{-1}$] | −7.85 (6) | −6.76 (6) | −5.7 (4) |
| *Climatic variables* | | | |
| SAT [°C] | 30.90 (7) | 21.56 (9) | 17.20 (9) |
| Temperature gradient [°C] | 18.3 (2) | 21.0 (2) | 24.3 (2) |
| NH temperature gradient [°C] | 17.2 (2) | 22.3 (2) | 28.5 (2) |
| SH temperature gradient [°C] | 19.4 (2) | 19.7 (3) | 20.0 (3) |
| Ocean temperature [°C] | 16.9987 (5) | 7.145 (2) | 5.289 (2) |
| Precipitation [g m$^{-2}$ s$^{-1}$] | 0.0477 (2) | 0.0377 (2) | 0.0352 (1) |
| Evaporation [g m$^{-2}$ s$^{-1}$] | 0.0477 (2) | 0.0377 (2) | 0.0352 (1) |
| Sea ice extent [$10^6$ km$^2$] | 0.008 (9) | 5.3 (9) | 34.3 (4) |
| Sea ice thickness [m] | - | 0.079 (4) | 0.864 (5) |
| Latitude of sea ice boundary (NH) | - | 81 | 62 |



**Table 4.** Convergence criteria of the self-consistent offline coupling iterations between BIOME4 and MITgcm. $S_{var}$ corresponds to the surface of land for which albedo is different compared to the previous iteration, rounded to a percent. SAT is averaged over 300 yr and error is the inter-annual variability. Bold values highlight satisfied convergence criteria. $B_{surf,o}$ is the global energy budget over ocean.

|  | Hot state | Warm state | Cold state |
|---|---|---|---|
| *Initial state* | | | |
| SAT [°C] | 30.92 (9) | 21.5 (1) | 17.1 (1) |
| $B_{surf,o}$ [W m$^{-2}$] | 0.1 (2) | −0.1 (3) | −0.1 (3) |
| | | | |
| *Iteration 1* | | | |
| $S_{var}$ [%] | 87 | 87 | 87 |
| SAT [°C] | 32.6 (1) | 21.2 (1) | 17.1 (1) |
| $B_{surf,o}$ [W m$^{-2}$] | 0.8 (2) | −0.2 (3) | 0.0 (3) |
| | | | |
| *Iteration 2* | | | |
| $S_{var}$ [%] | 37 | 12 | 15 |
| SAT [°C] | 32.3 (1) | 21.76 (9) | 18.0 (1) |
| $B_{surf,o}$ [W m$^{-2}$] | 0.7 (2) | 0.0 (3) | 0.1 (3) |
| | | | |
| *Iteration 3* | | | |
| $S_{var}$ [%] | 15 | **8** | 13 |
| SAT [°C] | **32.4 (1)** | **21.8 (1)** | **18.2 (1)** |
| $B_{surf,o}$ [W m$^{-2}$] | 0.8 (2) | 0.0 (3) | 0.2 (3) |
| | | | |
| *Iteration 4* | | | |
| $S_{var}$ [%] | 11 | - | **6** |
| SAT [°C] | **32.3 (1)** | - | **18.03 (9)** |
| $B_{surf,o}$ [W m$^{-2}$] | 0.8 (2) | - | 0.1 (3) |
| | | | |
| *Iteration 5* | | | |
| $S_{var}$ [%] | **8** | - | - |
| SAT [°C] | **32.4 (1)** | - | - |
| $B_{surf,o}$ [W m$^{-2}$] | 0.8 (2) | - | - |



**Table 5.** Carbon reservoirs and air-sea fluxes for the three attractors with initial atmospheric $pCO_2$ of 320 ppm and at the edges of stable branches.

| Initial fixed $pCO_2$ value | Equilibrium SAT with fixed $CO_2$ [°C] | Equilibrium SAT with variable $CO_2$ [°C] | Atmospheric $pCO_2$ [ppm] | Oceanic carbon [$10^{18}$ mol] | Total carbon [$10^{18}$ mol] | Air-sea carbon flux [ppm yr$^{-1}$] |
|---|---|---|---|---|---|---|
| *Hot state* | | | | | | |
| 308 ppm | 29.85 (8) | 29.7 (1) | 307.8 (3) | 2.7575 (1) | 2.8120 (2) | 0.0 (1) |
| 320 ppm | 30.90 (7) | 30.4 (9) | 312.2 (1) | 2.7600 (2) | 2.8120 (2) | 0.00 (5) |
| 440 ppm | > 37 | - | - | - | - | - |
| *Warm state* | | | | | | |
| 320 ppm | 21.56 (9) | 21.8 (1) | 325.1 (4) | 2.9018 (2) | 2.9594 (1) | -0.01 (4) |
| 328 ppm | 23.01 (8) | 22.57 (9) | 317.2 (3) | 2.9051 (1) | 2.9612 (1) | 0.01 (4) |
| *Cold state* | | | | | | |
| 304 ppm | 15.3 (1) | 15.7 (1) | 310.7 (3) | 2.9333 (2) | 2.9883 (1) | -0.0 (2) |
| 320 ppm | 17.20 (9) | 17.26 (9) | 320.8 (1) | 2.9456 (1) | 3.0024 (1) | 0.00 (4) |
| 332 ppm | 18.4 (1) | 17.72 (8) | 326.8 (2) | 2.94673 (8) | 3.0046 (1) | 0.01 (4) |



**Table C1.** Correspondence between the biomes used in BIOME4, ecosystem types and biomass density.

| Number and Name | | Ecosystem type | Biomass Density [Mg ha$^{-1}$] |
|---|---|---|---|
| 1 | Tropical evergreen broadleaf forest | | |
| 2 | Tropical semi-evergreen broadleaf forest | Tropical forests | 390 |
| 3 | Tropical deciduous broadleaf forest and woodland | | |
| 4 | Temperate deciduous broadleaf forest | | |
| 5 | Temperate evergreen needleleaf forest | Temperate forests | 270 |
| 6 | Warm-temperate evergreen broadleaf and mixed forest | | |
| 7 | Cool mixed forest | Temperate and boreal forests | 160 |
| 9 | Cool-temperate evergreen needleleaf and mixed forest | | |
| 8 | Cool evergreen needleleaf forest | | |
| 10 | Cold evergreen needleleaf forest | Boreal forests | 83 |
| 11 | Cold deciduous forest | | |
| 17 | Temperate evergreen needleleaf open woodland | | |
| 12 | Tropical savanna | Tropical savannas and grasslands | 57 |
| 19 | Tropical grassland | | |
| 13 | Tropical xerophytic shrubland | | |
| 14 | Temperate xerophytic shrubland | Mediterranean shrublands | 120 |
| 15 | Temperate sclerophyll woodland and shrubland | | |
| 20 | Temperate grassland | Temperate grasslands | 8 |
| 21 | Desert | Deserts | 7 |
| 27 | Barren | | |
| 22 | Graminoid and forb tundra | | |
| 23 | Low and high shrub tundra | | |
| 24 | Erect dwarf-shrub tundra | Arctic tundra | 7 |
| 25 | Prostrate dwarf-shrub tundra | | |
| 26 | Cushion-forb tundra | | |