# Peer review of "Alternative climatic steady states for the Permian-Triassic paleogeography"

_EGUsphere, 2023_

## Author Comment (AC1)

Manuscript: egusphere-2023-1808

**Title:** Alternative climatic steady states for the Permian-Triassic paleogeography

**Reviewer #1:**

Ragon et al. spent lots of effort in finding all the possible steady states for the Permian-Triassic paleo-geography using a relatively sophisticated Earth system model. The interesting finding, in my opinion, is that they found a 'warm' state in between the 'cold' and 'hot' states. This warm state cannot be reached from the either the cold or hot state by increasing or decreasing greenhouse gas forcings. The other findings are less interesting but worth being published on the journal EGUsphere.

We are thankful for this positive evaluation of our paper. We agree that the original part of our paper is the approach and the fact that multiple alternative steady states can be found for the same boundary conditions. We note however that we did not look for 'all the possible steady states', but that we restricted our analysis to those of relevance to the Early Triassic.

The major reason that the results may be less important than they seem to is that the multiple steady states found here could disappear when a fully coupled state-of-the-art climate model is used. To my own experience, the cold and hot climate states (even the cold state is still quite warm) presented in this manuscript could not coexist at the same forcing in the NCAR model family. Especially, the sea ice used in the model of this manuscript is thermodynamic only. When the NCAR model was run in such mode, a so-called Jormungand state could be found (Abbot et al., 2011; https://agupubs.onlinelibrary.wiley.com/doi/full/10.1029/2011JD015927) but never found in the fully coupled mode. Therefore, I think the authors should point out in the abstract or conclusion that the multiple steady states found in their study may depend on the specific configuration of their model, especially the neglect of sea-ice dynamics.

We agree that the number of steady states may depend on the configuration and the model setup. Indeed, feedback mechanisms acting on the same time scale could affect the balance between different processes and the number of steady states. A sentence on this aspect can be added in the Conclusions, mentioning the role of sea ice dynamics, and also ocean dynamics, and relevant references (at line 300):

*In particular, including sea ice dynamics or different numerical implementations of thermodynamic sea ice Lewis et al. (2007); Voigt & Abbot (2012); Hörner & Voigt (2023), as well as considering a mixed-layer ocean or a fully dynamical one Poulsen et al. (2001); Pohl et al. (2014), may change the number of steady states, and reveal the source of possible biases.*

This is also the reason why in the Conclusions (last paragraph, line 299) we call for the need of comparing different climate models. We can also specify that we use a *thermodynamic* sea ice in the abstract and the Conclusions (second paragraph, line 278) when we mention the MITgcm configuration.

Moreover, I think a snowball Earth branch should exist in their model if the initial condition is cold enough. They do not need to explore the full branch but just confirming their existence is necessary in this kind of study.

As mentioned above, we did not look for all the possible steady states. However, we indeed confirm that at least a colder climate exists: at the lower edge of the stable branches of both hot and cold states the system is attracted towards a colder state, as shown in Fig. B1 of the manuscript. A waterbelt state is present where the ice extends to $\sim 30°$ latitude and the global mean surface air temperature is approx. $-10\,°C$ (see Fig. 1 in this response). However, we have not investigated this state further since simulations require long CPU time while geological data exclude for the presence of snowball or waterbelt states in the Early Triassic Sun et al. (2012); Romano et al. (2013); Goudemand et al. (2019); Widmann et al. (2020).

We propose to include the following paragraph in the Introduction (at line 40) to describe the climatic oscillations in the aftermath of the Permian-Triassic Boundary mass extinction and to provide the general context for our numerical simulations.

*We are interested in the climatic oscillations observed at the Smithian-Spathian boundary in the Early Triassic, just after the Permian-Triassic boundary (PTB) mass extinction ($\sim 252$ Ma), the most severe of the Phanerozoic Raup (1979); MacLeod (2014); Stanley (2016). As a consequence of the volcanic activity of the Siberian Large Igneous Province Campbell et al. (1992); Renne et al. (1995); Reichow et al. (2009), the global carbon cycle entered a perturbed state which persisted for nearly 5.4 Myr in the Early Triassic, until a new equilibrium state was reached in the Anisian Sun et al. (2012); Romano et al. (2013); Goudemand et al. (2019); Leu et al. (2019); Widmann et al. (2020). The observed fluctuations in the carbon isotope record Payne et al. (2004); Galfetti et al. (2007); Retallack et al. (2011) with successive diversification-extinction cycles of the nekton Orchard (2007); Brühwiler et al. (2010); Leu et al. (2019)*

[Figure]

Figure 1: Sea-surface temperature and sea-ice extent in waterbelt.

*and ecological crises of terrestrial plants Hermann et al. (2011); Schneebeli-Hermann (2020), are all indicative of the climatic changes which occurred in the Early Triassic, with variations in the global temperature of the order of 7-8 °C from the thermal maximum in the late Smithian to cold climates in early Spathian times Widmann et al. (2020).*

*However, despite the great effort of the scientific community to reconstruct such climatic oscillations in the aftermath of the PTB, large uncertainties remain in timing and causal relationships, implying that numerical modelling needs to consider a wide range of initial and boundary conditions for the simulation of such geological interval. In this context, ...* [continuing at line 44]

L164: "stronger" than what? This makes the sentence hard to understand.

We meant 'stronger than the heat transport in a climate without sea ice'. We suggest to reformulate this sentence as follows:

*As reported in an aquaplanet configuration Ragon et al. (2022), the presence of sea ice increases the meridional surface air temperature gradient, defined as the temperature difference between polar (30° to 90°) and equatorial (-30° to 30°) regions, and thus makes the heat transport stronger than for an ice-free climate.*

Fig. 6: It has been pointed in the literature that the strength of annual mean Hadley circulation may not be meaningful, can the authors please confirm that the strength of seasonal Hadley circulation has a similar trend?

We thank the reviewer for pointing this out. Like in the present-day climate, the annual mean Hadley cells are quite similar to the cells during spring and autumn seasons, while in the winter season, the circulation is dominated by a single cell, the one of the winter hemisphere Lindzen & Hou (1988); Dima & Wallace (2003). Thus, we agree that it is useful to show the winter Hadley circulation, for example in a new Appendix (see Fig. 2 in this response). The following text can be added at line 176, section 3.1:

*In the winter season, the Hadley circulation is dominated by a single cell, the one of the winter hemisphere Lindzen & Hou (1988); Dima & Wallace (2003), as shown in Fig. (in the appendix). The austral winter cell in the hot state becomes weaker and slightly wider than that of the cold state, in agreement with the trend observed in simulations of the Pangea hot climate at ca. 250 Ma in comparison to pre-industrial conditions Zhang et al. (2023). The boreal winter cell has a similar width in the three attractors, while its intensity is maximal and more shifted towards the Equator in the cold state.*

These trends of the winter cells are thus also consistent with what is observed in the mean annual Hadley circulation in Fig. 6 in the manuscript.

Figs. 7 and 8: I think these two figures can be combined into one

[Figure]

Figure 2: Seasonal mean atmospheric meridional overturning circulation for hot state (a,d), warm state (b,e) and cold state (c,f). Panels (a-c) and (d-f) corresponds to summer (June-July-August, JJA) and winter (December-January-February, DJF) seasons of the northern hemisphere, respectively.

We agree with this proposition.

**References**

Brühwiler, T., Bucher, H., Brayard, A., & Goudemand, N. (2010). High-resolution biochronology and diversity dynamics of the early triassic ammonoid recovery: The smithian faunas of the northern indian margin. *Palaeogeography, Palaeoclimatology, Palaeoecology*, *297*(2), 491–501.
URL https://www.sciencedirect.com/science/article/pii/S0031018210005274

Campbell, I. H., Czamanske, G. K., Fedorenko, V. A., Hill, R. I., & Stepanov, V. (1992). Synchronism of the siberian traps and the permian-triassic boundary. *Science*, *258*(5089), 1760–1763.
URL https://www.science.org/doi/abs/10.1126/science.258.5089.1760

Dima, I. M., & Wallace, J. M. (2003). On the seasonality of the hadley cell. *Journal of the Atmospheric Sciences*, *60*(12), 1522 – 1527.
URL https://journals.ametsoc.org/view/journals/atsc/60/12/1520-0469_2003_060_1522_otsoth_2.0.co_2.xml

Galfetti, T., Bucher, H., Ovtcharova, M., Schaltegger, U., Brayard, A., Brühwiler, T., Goudemand, N., Weissert, H., Hochuli, P. A., Cordey, F., & Guodun, K. (2007). Timing of the early triassic carbon cycle perturbations inferred from new u–pb ages and ammonoid biochronozones. *Earth and Planetary Science Letters*, *258*(3), 593–604.
URL https://www.sciencedirect.com/science/article/pii/S0012821X0700252X

Goudemand, N., Romano, C., Leu, M., Bucher, H., Trotter, J. A., & Williams, I. S. (2019). Dynamic interplay between climate and marine biodiversity upheavals during the early triassic smithian -spathian biotic crisis. *Earth-Science Reviews*, *195*, 169–178.
URL https://www.sciencedirect.com/science/article/pii/S0012825218302563

Hermann, E., Hochuli, P. A., Bucher, H., Brühwiler, T., Hautmann, M., Ware, D., & Roohi, G. (2011). Terrestrial ecosystems on north gondwana following the end-permian mass extinction. *Gondwana Research*, *20*(2), 630–637.
URL https://www.sciencedirect.com/science/article/pii/S1342937X1100030X

Hörner, J., & Voigt, A. (2023). Sea-ice thermodynamics can determine waterbelt scenarios for snowball earth. *EGUsphere*, *2023*, 1–13.
URL https://egusphere.copernicus.org/preprints/2023/egusphere-2023-2073/

Leu, M., Bucher, H., & Goudemand, N. (2019). Clade-dependent size response of conodonts to environmental changes during the late smithian extinction. *Earth-Science Reviews*, *195*, 52–67.
URL https://www.sciencedirect.com/science/article/pii/S0012825218302216

Lewis, J. P., Weaver, A. J., & Eby, M. (2007). Snowball versus slushball earth: Dynamic versus nondynamic sea ice? *Journal of Geophysical Research: Oceans*, *112*(C11).
URL https://agupubs.onlinelibrary.wiley.com/doi/abs/10.1029/2006JC004037

Lindzen, R. S., & Hou, A. V. (1988). Hadley circulations for zonally averaged heating centered off the equator. *Journal of Atmospheric Sciences*, *45*(17), 2416 – 2427.
URL https://journals.ametsoc.org/view/journals/atsc/45/17/1520-0469_1988_045_2416_hcfzah_2_0_co_2.xml

MacLeod, N. (2014). The geological extinction record: History, data, biases, and testing. In *Volcanism, Impacts, and Mass Extinctions: Causes and Effects*. Geological Society of America.
URL https://doi.org/10.1130/2014.2505(01)

Orchard, M. J. (2007). Conodont diversity and evolution through the latest permian and early triassic upheavals. *Palaeogeography, Palaeoclimatology, Palaeoecology*, *252*(1), 93–117.
URL https://www.sciencedirect.com/science/article/pii/S0031018207001113

Payne, J. L., Lehrmann, D. J., Wei, J., Orchard, M. J., Schrag, D. P., & Knoll, A. H. (2004). Large perturbations of the carbon cycle during recovery from the end-permian extinction. *Science*, *305*(5683), 506–509.
URL https://www.science.org/doi/abs/10.1126/science.1097023

Pohl, A., Donnadieu, Y., Le Hir, G., Buoncristiani, J.-F., & Vennin, E. (2014). Effect of the ordovician paleogeography on the (in)stability of the climate. *Climate of the Past*, *10*(6), 2053–2066.
URL https://cp.copernicus.org/articles/10/2053/2014/

Poulsen, C. J., Pierrehumbert, R. T., & Jacob, R. L. (2001). Impact of ocean dynamics on the simulation of the neoproterozoic "snowball earth". *Geophysical Research Letters*, *28*(8), 1575–1578.
URL https://agupubs.onlinelibrary.wiley.com/doi/abs/10.1029/2000GL012058

Ragon, C., Lembo, V., Lucarini, V., Vérard, C., Kasparian, J., & Brunetti, M. (2022). Robustness of competing climatic states. *Journal of Climate*, *35*(9), 2769 – 2784.
URL https://journals.ametsoc.org/view/journals/clim/35/9/JCLI-D-21-0148.1.xml

Raup, D. M. (1979). Size of the permo-triassic bottleneck and its evolutionary implications. *Science*, *206*(4415), 217–218.
URL https://www.science.org/doi/abs/10.1126/science.206.4415.217

Reichow, M. K., Pringle, M., Al'Mukhamedov, A., Allen, M., Andreichev, V., Buslov, M., Davies, C., Fedoseev, G., Fitton, J., Inger, S., Medvedev, A., Mitchell, C., Puchkov, V., Safonova, I., Scott, R., & Saunders, A. (2009). The timing and extent of the eruption of the siberian traps large igneous province: Implications for the end-permian environmental crisis. *Earth and Planetary Science Letters*, *277*(1), 9–20.
URL https://www.sciencedirect.com/science/article/pii/S0012821X08006389

Renne, P. R., Black, M. T., Zichao, Z., Richards, M. A., & Basu, A. R. (1995). Synchrony and causal relations between permian-triassic boundary crises and siberian flood volcanism. *Science*, *269*(5229), 1413–1416.
URL https://www.science.org/doi/abs/10.1126/science.269.5229.1413

Retallack, G. J., Sheldon, N. D., Carr, P. F., Fanning, M., Thompson, C. A., Williams, M. L., Jones, B. G., & Hutton, A. (2011). Multiple early triassic greenhouse crises impeded recovery from late permian mass extinction. *Palaeogeography, Palaeoclimatology, Palaeoecology*, *308*(1), 233–251.
URL https://www.sciencedirect.com/science/article/pii/S0031018210005924

Romano, C., Goudemand, N., Vennemann, T. W., Ware, D., Schneebeli-Hermann, E., Hochuli, P. A., Brühwiler, T., Brinkmann, W., & Bucher, H. (2013). Climatic and biotic upheavals following the end-permian mass extinction. *Nature Geoscience*, *6*(1), 57–60.

Schneebeli-Hermann, E. (2020). Regime shifts in an early triassic subtropical ecosystem. *Frontiers in Earth Science*, *8*.
URL https://www.frontiersin.org/articles/10.3389/feart.2020.588696

Stanley, S. M. (2016). Estimates of the magnitudes of major marine mass extinctions in earth history. *Proceedings of the National Academy of Sciences*, *113*(42), E6325–E6334.
URL https://www.pnas.org/doi/abs/10.1073/pnas.1613094113

Sun, Y., Joachimski, M. M., Wignall, P. B., Yan, C., Chen, Y., Jiang, H., Wang, L., & Lai, X. (2012). Lethally hot temperatures during the early triassic greenhouse. *Science*, *338*(6105), 366–370.
URL https://www.science.org/doi/abs/10.1126/science.1224126

Voigt, A., & Abbot, D. S. (2012). Sea-ice dynamics strongly promote snowball earth initiation and destabilize tropical sea-ice margins. *Climate of the Past*, *8*(6), 2079–2092.
URL https://cp.copernicus.org/articles/8/2079/2012/

Widmann, P., Bucher, H., Leu, M., Vennemann, T., Bagherpour, B., Schneebeli-Hermann, E., Goudemand, N., & Schaltegger, U. (2020). Dynamics of the largest carbon isotope excursion during the early triassic biotic recovery. *Frontiers in Earth Science*, *8*.
URL https://www.frontiersin.org/articles/10.3389/feart.2020.00196

Zhang, S., Hu, Y., Yang, J., Li, X., Kang, W., Zhang, J., Liu, Y., & Nie, J. (2023). The hadley circulation in the pangea era. *Science Bulletin*, *68*(10), 1060–1068.
URL https://www.sciencedirect.com/science/article/pii/S209592732300275X

---

## Author Comment (AC2)

Manuscript: egusphere-2023-1808

**Title:** Alternative climatic steady states for the Permian-Triassic paleogeography

**Reviewer #2:**

Ragon and others explore climate of the Permian-Triassic with the MITGCM. They find three equilibrium climate states with an initial $CO_2$ of 320 ppm. From these initial states, they explore the range of stability and the importance of vegetation and carbon cycle feedbacks.

I found this to be an interesting study. The experiment design was well thought out. The text and figures were easy to follow. Here are some comments that I think could improve the manuscript.

We are grateful for this positive evaluation of our study.

A lot of steps went into the model spin up. They are well documented in the text, which I appreciate. Although the text is generally clear, I think a schematic of the spin up procedure would help the reader understand what was done.

We thank the reviewer for this suggestion. We can add the following schematic for the whole procedure in a new appendix (or in Supplementary Material).

[Figure]

Figure 1: Schematic of the numerical procedure.

There was a lot of work put into creating realistic boundary conditions. This is worthwhile since most previous MITGCM multiple equilibria studies used idealized configurations. One of the benefits of realistic boundary conditions is model-proxy comparison. The authors speculate that the mismatch between HadCM3 simulations and a proxy reconstruction might be the result of multiple equilibria. However, the authors do not make any model-proxy comparisons themselves. I do not think many temperature records of the Permian-Triassic exist, but it is still worth doing.

We suggest to add a new paragraph in the Introduction (at line 40) to better explain the context of our numerical simulations:

*We are interested in the climatic oscillations observed at the Smithian-Spathian boundary in the Early Triassic, just after the Permian-Triassic boundary (PTB) mass extinction (∼252 Ma), the most severe of the Phanerozoic (Raup, 1979; MacLeod, 2014; Stanley, 2016). As a consequence of the volcanic activity of the Siberian Large Igneous Province (Campbell et al., 1992; Renne et al., 1995; Reichow et al., 2009), the global carbon cycle entered a perturbed state which persisted for nearly 5.4 Myr in the Early Triassic, until a new equilibrium state was reached in the Anisian (Sun et al., 2012; Romano et al., 2013; Goudemand et al., 2019; Leu et al., 2019; Widmann et al., 2020). The observed fluctuations in the carbon isotope*

*record (Payne et al., 2004; Galfetti et al., 2007; Retallack et al., 2011) with successive diversification-extinction cycles of the nekton (Orchard, 2007; Brühwiler et al., 2010; Leu et al., 2019) and ecological crises of terrestrial plants (Hermann et al., 2011; Schneebeli-Hermann, 2020), are all indicative of the climatic changes which occurred in the Early Triassic, with variations in the global temperature of the order of 7-8 °C from the thermal maximum in the late Smithian to cold climates in early Spathian times (Widmann et al., 2020).*
*However, despite the great effort of the scientific community to reconstruct such climatic oscillations in the aftermath of the PTB, large uncertainties remain in timing and causal relationships, implying that numerical modelling needs to consider a wide range of initial and boundary conditions for the simulation of such geological interval. In this context, ...* [continuing at line 44]

A detailed comparison between biomes and plant fossil data goes well beyond the scope of the present work. We are preparing such comparison for a forthcoming publication. The aim of the present work is to focus on the climatic states that we obtain with the same paleogeography and different initial conditions, in order to open the possibility of using alternative steady states and tipping mechanisms in future proxy comparisons for the Early Triassic.

On the point of realism, the range of $CO_2$ for these equilibrium states is quite small from a geologic perspective. What are the implications of this?

The atmospheric module is based on SPEEDY, the Simplified Parametrizations primitivE-Equation DYnamics described in Molteni (2003) (see also the Appendix to that article on the web page of SPEEDY). In the parameterization scheme for the longwave radiation, the infrared spectrum is partitioned into four regions: 1) the 'infrared window' between 8.5 and 11 $\mu$m; 2) the band of strong absorption by $CO_2$ around 15 $\mu$m; 3) the aggregation of regions with weak/moderate absorption by water vapour; 4) the aggregation of regions with strong absorption by water vapour. Thus, the absorption of $CO_2$ is limited by the fact that only the largest absorption band is included in the infrared spectrum. This implies that the atmospheric $CO_2$ content is probably underestimated in our simulations, and that the range of $CO_2$ for the steady states can change when considering the full absorption bands. We propose to include this more detailed description of the infrared spectrum in SPEEDY in Section 2.1.

Another aspect to consider is that the various estimates for $pCO_2$ content around the Permian-Triassic Boundary and Early Triassic (e. g. Royer et al. (2004); Berner (2006); Foster et al. (2017); Lenton et al. (2018); Joachimski et al. (2022)) have relatively low values for the Phanerozoic with large error bars. As mentioned in the manuscript at line 100, Fig. 1 (and corresponding data in Supplementary Material) in Foster et al. (2017) shows huge uncertainties for the Early Triassic, with $CO_2$ values ranging between 0 and 1800 ppm.

Also, a few sentences of discussion about the biomes of the Permian-Triassic versus the biomes in BIOME4 would be worthwhile.

BIOME4 is based on the concept of Plant Funcional Type (PFT), rather than taxonomic grouping (Kaplan, 2001). Despite the definition of PFTs is based on present-day plants, this is the only available framework that can be applied to plants at different deep-time geological periods, assuming that their functionality was similar to the present-day one. Within this framework, plants with certain fundamental characteristis (*i. e.*, growth form, phenology, rooting depth) are grouped together and included in BIOME4, allowing for studying their distribution at the global scale. BIOME4 has been applied to study the Jurassic climate (Sellwood & Valdes, 2008), the Middle Pliocene (3.6-2.6 Ma; Salzmann et al. (2008)), the Last Glacial Maximum (Harrison & Prentice, 2003). We suggest to include the concept of PFTs and relevant literature in Section 2.3.

As said above, we are presently comparing our simulation results with plant fossil data. For the forthcoming study, we will adapt the output of BIOME4 to biomes defined as in Ziegler (1990) and Nowak et al. (2020). Such definition is more general and applicable to the Permian-Triassic period. However, in the present study we prefer to consider the same definition of mega-biomes as in many other studies of the past, as mentioned in the previous paragraph.

Line 81-82: Can you provide a bit more information about how the model conserves energy?

The energy conservation of the MITgcm has been investigated by Campin et al. (2008) in coupled simulations. The accuracy to which heat is conserved (but also salt and fresh water) is limited by machine precision, with a time-stepping implementation which turns out to be stable and conservative. We have further investigated the energy conservation issue in our MITgcm setup in Brunetti & Vérard (2018); Brunetti et al. (2019); Ragon et al. (2022). As discussed in Lucarini & Ragone (2011), even if a model can perfectly conserve energy between its components at the ocean surface, as the MITgcm does, it can still have energy sources/sinks that affect the TOA budget, that are not accounted for and are sometimes

called 'ghost energy' (Pascale et al., 2012). As discussed in Lucarini & Ragone (2011) and Liepert & Previdi (2012), the spurious bias is due to physical processes that have been neglected, inconsistently treated, or approximated in climate models, as well as to unphysical effects of numerical dissipation (Pascale et al., 2012; Mauritsen et al., 2012; Lauritzen & Williamson, 2019; Trenberth, 2020). The TOA energy balance ranges between -0.2 and 4.8 W/m$^2$ in the preindustrial scenario with CMIP3 (as shown in Fig. 2A in Lucarini & Ragone (2011)), and between -3.16 and 2.37 in CMIP5 (see Table 2 in Lembo et al. (2019)). In our setting, the range is between -0.4 and -0.1 W/m$^2$ (see Table 3), and the ocean surface energy imbalance is less than 0.1 W/m$^2$ in absolute value, since we run the simulations towards equilibrium. The main contribution to TOA energy imbalance comes from the fact that sea ice dynamics is neglected in our setup (Brunetti & Vérard, 2018). We have discussed these conservation properties in our previous papers (Brunetti & Vérard, 2018; Brunetti et al., 2019; Ragon et al., 2022). We can refer to these papers in the revised manuscript and add some more information on energy conservation in our Permian-Triassic setup in Section 3.

Line 84: I do not think Foster et al. (2017) is the original source. Maybe Gough (1981)?

The original reference is indeed Gough (1981), and Foster et al. (2017) refer to this paper.

Line 106: Is it possible the tipping points could occur with less forcing if the simulations were run for more than 100 years?

The trade-off between the increase in forcing $\Delta S$ and the interval of time $\Delta T$ is discussed in Brunetti & Ragon (2023) (Method II). The change of the forcing $\Delta S$ should be sufficiently low to avoid rate-induced tipping (R-tipping). On the other hand, this method becomes interesting from the point of view of reducing the computational time if $\Delta T$ is not too large. We have tested different values in a coupled aquaplanet configuration, where the bifurcation diagram is obtained with large relaxation times (standard method in Brunetti & Ragon (2023)). We saw that taking $\Delta T \sim 100$ yr was sufficient to reconstruct well the position of tipping points. This is why we have used the same value here. If $\Delta T$ is increased, with $\Delta S$ sufficiently small to avoid R-tipping, the estimated position of the tipping point can slightly change, but the overall picture of the bifurcation diagram remains the same. The larger $\Delta T$, the more precise the position of tipping points, the higher the computational cost.

Line 119: Is there any transpiration component in the MITgcm?

Transpiration is not taken into account in the MITgcm atmospheric module. We can add this information in the model description (Section 3.1).

Line 123: "run BIOME4 again"

We thank the reviewer for pointing out this typo.

Line 143: 0.1 ppm per year seems like a large drift on geologic time scales. Does it look like the carbon is heading towards an equilibrium or continually drifting?

A value smaller than 0.1 ppm/yr is required only at the end of the first step. At the final step, the equilibrium values are reported in Table 5 and are consistent with a null flux (within uncertainties). We suggest to reformulate this sentence at step 1, the paragraph at step 2 and the footnote by avoiding the word 'equilibrium', which is misleading in this context.

Line 203: "to simulate larger pCO2 values"

This sentence will be corrected indeed as suggested.

Line 204: To clarify, the model boundary conditions are largely responsible for the high temperature sensitivity?

The reviewer is right: the boundary conditions (the Permian-Triassic paleogeography) have an impact on the temperature of the steady states, for the same $CO_2$ content. Using boundary conditions for an aquaplanet gives steady states with different global mean temperatures (Brunetti et al., 2019; Brunetti & Ragon, 2023). However, the numerical instability that we observe at mean global surface air temperatures larger than 38 °C does not depend on such boundary conditions, but on the atmospheric module of MITgcm (SPEEDY, based on simplified parametrizations described in Section 2.1; see also Molteni (2003)).

Line 212: Based on Figure B1, it does not seem like the cold state can be reached from the warm state, so the loop is not closed. What are the implications of this with respect to real climate evolution?

The cold state stability range extends beyond the warm state branch on both sides, as it can be seen in Fig. 11 where only stable regions have been shown. Thus, the cold state can be reached from the warm state through bifurcation-induced tipping (B-tipping). However, the warm state cannot be reached

from the cold state by B-tipping, as we say at lines 207-208. Implications are that the warm state can in principle be attained through other tipping mechanisms, as for example noise-induced tipping due to perturbations of the biological pump. As we state in the manuscript (lines 209-210), if we assume that the size of the warm state basin of attraction is proportional to the branch length, catching the warm attractor would require very specific initial conditions and is therefore quite unlikely to occur.

Line 216: I do not follow this argument. HadCM3, like other higher complexity models, does not produce multiple equilibria. CO2 and proxy reconstruction uncertainties seem more probably explanations for the mismatch. Are you arguing that HadCM3 is somehow stuck in a cold equilibrium solution? Also, is it OK to cite an EGU presentation?

We disagree with the reviewer, since nowadays several high compexity models show mutiple equilibria. For example, the eddy-permitting GCM HadGEM3-GC2 shows hysteresis of the Atlantic Meridional Over-turning Circulation over centennial time scales (Jackson & Wood, 2018). Similarly, Peltier & Vettoretti (2014) show abrupt transitions in CESM1 over millennial time scales. A result that has been repeated by a number of other IPCC-class models (see review by Malmierca-Vallet et al. (2023)), including HadCM3B. Hence higher complexity models are indeed able to produce multiple equilibria.

On the discrepancy between simulation results and proxy data, we discussed at EGU2023 with Prof. Paul Valdes, who is now interested in performing additional numerical simulations with HadCM3 to explore the hypothesis of the presence of multiple steady states under the same Permian-Triassic boundary conditions. Unfortunely, he has not had time until now to publish a paper about that, thus we can only cite his presentation at EGU, which anyway has a doi number. We can also add: 'Paul Valdes, *personal communication*'.

Table 3: I am confused by the differences in energy balance between the ocean surface and TOA. Where is the energy going?

See the point above on the energy conservation (lines 81-82). The surface energy balance becomes nearly zero if simulations are long enough, as in our case where the simulations are run towards equilibrium. The TOA energy balance can be affected by the so-called 'ghost energy', mainly due to numerical dissipation or missed processes in the simulation setup (Lucarini & Ragone, 2011; Brunetti & Vérard, 2018). In our simulations, the TOA budget ranges between -0.4 and -0.1 W/m2 (see Table 3), which are small values in comparison to those obtained by some CMIP models (see references above).

Some of the figures would be easier to interpret with difference plots. I think figures 7, 8, and 9 in particular.

We thank the reviewer for this suggestion. Difference plots indeed improve the comparison of precipitation and evaporation, especially in warm and cold states where these quantities are quite similar, as shown below (Fig. 2 in this response).
Regarding the salinity field, we think that difference plots are not useful (see Fig. 3 below), since we are interested to show that the salinity distribution is symmetrical in the hot state with respect to the Equator because of the absence of ice. Thus, we prefer not to use difference plots for the salinity. Instead, contour lines can be added to improve readability (Fig. 4 below).

Figure D1: I do not understand the temperature response with the carbon cycle. Why does the sensitivity change, especially in the warm state? Also, I am surprised that turning on the carbon cycle did not lead to large changes in atmospheric CO2. Any explanation?

The stable branch of the warm state is very short, and thus a small perturbation in the forcing can easily induce a loss of the steady state. This indeed is what happens by including air-sea carbon exchanges. The warm state looses stability on the (short) branch. In contrast, the climate sensitivity of the other two steady states does not change much when air-sea carbon exchanges are allowed.
The fact that allowing carbon exchanges only slightly affects the atmospheric $CO_2$ content (and the equilibrium temperature) is due to the procedure used, which is described in section 2.3. We first estimate the distributions of oceanic tracers (DIC, dissolved inorganic phosphorous, alkalinity, phosphate and oxygen), which correspond to given values of atmospheric $CO_2$ content on the stable branch of the attractors (step 1). Then, these distributions are used when the air-sea carbon exchange is allowed (step 2). By construction, air-sea carbon exchanges vary the atmospheric $CO_2$ content of the order of 10 ppm (see Table 5). The advantage of including the option of $CO_2$ exchanges is to obtain the total (ocean + atmosphere) carbon content for each state, as reported in Table 5.

[Figure]

Figure 2: Precipitation (up) and evaporation (bottom) for *(a, d)*: hot - cold; *(b, e)*: warm - cold; *(c, f)* cold).

[Figure]

Figure 3: Difference plots for the salinity field: *(a)* hot - cold; *(b)* warm - cold; *(c)* cold.

[Figure]

Figure 4: Salinity field for *(a)* hot; *(b)* warm; *(c)* cold states.

**References**

Berner, R. A. (2006). Geocarbsulf: A combined model for phanerozoic atmospheric o2 and co2. *Geochimica et Cosmochimica Acta*, *70*(23), 5653–5664. A Special Issue Dedicated to Robert A. Berner. URL https://www.sciencedirect.com/science/article/pii/S0016703706002031

Brunetti, M., Kasparian, J., & Vérard, C. (2019). Co-existing climate attractors in a coupled aquaplanet.

*Climate Dynamics*, *53*(9-10), 6293–6308.
URL http://link.springer.com/10.1007/s00382-019-04926-7

Brunetti, M., & Ragon, C. (2023). Attractors and bifurcation diagrams in complex climate models. *Phys. Rev. E*, *107*, 054214.
URL https://link.aps.org/doi/10.1103/PhysRevE.107.054214

Brunetti, M., & Vérard, C. (2018). How to reduce long-term drift in present-day and deep-time simulations? *Climate dynamics*, *50*(11-12), 4425–4436.

Brühwiler, T., Bucher, H., Brayard, A., & Goudemand, N. (2010). High-resolution biochronology and diversity dynamics of the early triassic ammonoid recovery: The smithian faunas of the northern indian margin. *Palaeogeography, Palaeoclimatology, Palaeoecology*, *297*(2), 491–501.
URL https://www.sciencedirect.com/science/article/pii/S0031018210005274

Campbell, I. H., Czamanske, G. K., Fedorenko, V. A., Hill, R. I., & Stepanov, V. (1992). Synchronism of the siberian traps and the permian-triassic boundary. *Science*, *258*(5089), 1760–1763.
URL https://www.science.org/doi/abs/10.1126/science.258.5089.1760

Campin, J.-M., Marshall, J., & Ferreira, D. (2008). Sea ice–ocean coupling using a rescaled vertical coordinate z*. *Ocean Modelling*, *24*(1), 1–14.
URL https://www.sciencedirect.com/science/article/pii/S1463500308000553

Foster, G., Royer, D., & Lunt, D. (2017). Future climate forcing potentially without precedent in the last 420 million years. *Nature Communications 8*, *14845*.

Galfetti, T., Bucher, H., Ovtcharova, M., Schaltegger, U., Brayard, A., Brühwiler, T., Goudemand, N., Weissert, H., Hochuli, P. A., Cordey, F., & Guodun, K. (2007). Timing of the early triassic carbon cycle perturbations inferred from new u–pb ages and ammonoid biochronozones. *Earth and Planetary Science Letters*, *258*(3), 593–604.
URL https://www.sciencedirect.com/science/article/pii/S0012821X0700252X

Goudemand, N., Romano, C., Leu, M., Bucher, H., Trotter, J. A., & Williams, I. S. (2019). Dynamic interplay between climate and marine biodiversity upheavals during the early triassic smithian -spathian biotic crisis. *Earth-Science Reviews*, *195*, 169–178.
URL https://www.sciencedirect.com/science/article/pii/S0012825218302563

Gough, D. O. (1981). Solar interior structure and luminosity variations. In V. Domingo (Ed.) *Physics of Solar Variations*, (pp. 21–34). Dordrecht: Springer Netherlands.

Harrison, S. P., & Prentice, C. I. (2003). Climate and co2 controls on global vegetation distribution at the last glacial maximum: analysis based on palaeovegetation data, biome modelling and palaeoclimate simulations. *Global Change Biology*, *9*(7), 983–1004.

Hermann, E., Hochuli, P. A., Bucher, H., Brühwiler, T., Hautmann, M., Ware, D., & Roohi, G. (2011). Terrestrial ecosystems on north gondwana following the end-permian mass extinction. *Gondwana Research*, *20*(2), 630–637.
URL https://www.sciencedirect.com/science/article/pii/S1342937X1100030X

Jackson, L. C., & Wood, R. A. (2018). Hysteresis and resilience of the amoc in an eddy-permitting gcm. *Geophysical Research Letters*, *45*(16), 8547–8556.
URL https://agupubs.onlinelibrary.wiley.com/doi/abs/10.1029/2018GL078104

Joachimski, M. M., Müller, J., Gallagher, T. M., Mathes, G., Chu, D. L., Mouraviev, F., Silantiev, V., Sun, Y. D., & Tong, J. N. (2022). Five million years of high atmospheric CO2 in the aftermath of the Permian-Triassic mass extinction. *Geology*, *50*(6), 650–654.
URL https://doi.org/10.1130/G49714.1

Kaplan, J. O. (2001). Geophysical applications of vegetation modeling. Tech. rep., Lund University.

Lauritzen, P. H., & Williamson, D. L. (2019). A total energy error analysis of dynamical cores and physics-dynamics coupling in the community atmosphere model (cam). *Journal of Advances in Modeling Earth Systems*, *11*(5), 1309–1328.
URL https://agupubs.onlinelibrary.wiley.com/doi/abs/10.1029/2018MS001549

Lembo, V., Lunkeit, F., & Lucarini, V. (2019). Thediato (v1.0) – a new diagnostic tool for water, energy and entropy budgets in climate models. *Geoscientific Model Development*, *12*(8), 3805–3834.
URL https://gmd.copernicus.org/articles/12/3805/2019/

Lenton, T. M., Daines, S. J., & Mills, B. J. (2018). Copse reloaded: An improved model of biogeochemical cycling over phanerozoic time. *Earth-Science Reviews*, *178*, 1–28.
URL https://www.sciencedirect.com/science/article/pii/S0012825217304117

Leu, M., Bucher, H., & Goudemand, N. (2019). Clade-dependent size response of conodonts to environmental changes during the late smithian extinction. *Earth-Science Reviews*, *195*, 52–67.
URL https://www.sciencedirect.com/science/article/pii/S0012825218302216

Liepert, B. G., & Previdi, M. (2012). Inter-model variability and biases of the global water cycle in cmip3 coupled climate models. *Environmental Research Letters*, *7*(1), 014006.

Lucarini, V., & Ragone, F. (2011). Energetics of climate models: Net energy balance and meridional enthalpy transport. *Reviews of Geophysics*, *49*(1), RG1001.
URL https://agupubs.onlinelibrary.wiley.com/doi/abs/10.1029/2009RG000323

MacLeod, N. (2014). The geological extinction record: History, data, biases, and testing. In *Volcanism, Impacts, and Mass Extinctions: Causes and Effects*. Geological Society of America.
URL https://doi.org/10.1130/2014.2505(01)

Malmierca-Vallet, I., Sime, L. C., & the D-O community members (2023). Dansgaard–oeschger events in climate models: review and baseline marine isotope stage 3 (mis3) protocol. *Climate of the Past*, *19*(5), 915–942.
URL https://cp.copernicus.org/articles/19/915/2023/

Mauritsen, T., Stevens, B., Roeckner, E., Crueger, T., Esch, M., Giorgetta, M., Haak, H., Jungclaus, J., Klocke, D., Matei, D., Mikolajewicz, U., Notz, D., Pincus, R., Schmidt, H., & Tomassini, L. (2012). Tuning the climate of a global model. *Journal of Advances in Modeling Earth Systems*, *4*, M00A01.

Molteni, F. (2003). Atmospheric simulations using a GCM with simplified physical parametrizations. I: model climatology and variability in multi-decadal experiments. *Climate Dynamics*, *20*(2), 175–191.
URL http://link.springer.com/10.1007/s00382-002-0268-2

Nowak, H., Vérard, C., & Kustatscher, E. (2020). Palaeophytogeographical patterns across the permian–triassic boundary. *Frontiers in Earth Science*, *8*, 613350.

Orchard, M. J. (2007). Conodont diversity and evolution through the latest permian and early triassic upheavals. *Palaeogeography, Palaeoclimatology, Palaeoecology*, *252*(1), 93–117.
URL https://www.sciencedirect.com/science/article/pii/S0031018207001113

Pascale, S., Gregory, J. M., Ambaum, M. H., & Tailleux, R. (2012). A parametric sensitivity study of entropy production and kinetic energy dissipation using the famous aogcm. *Climate dynamics*, *38*(5-6), 1211–1227.

Payne, J. L., Lehrmann, D. J., Wei, J., Orchard, M. J., Schrag, D. P., & Knoll, A. H. (2004). Large perturbations of the carbon cycle during recovery from the end-permian extinction. *Science*, *305*(5683), 506–509.
URL https://www.science.org/doi/abs/10.1126/science.1097023

Peltier, W. R., & Vettoretti, G. (2014). Dansgaard-oeschger oscillations predicted in a comprehensive model of glacial climate: A "kicked" salt oscillator in the atlantic. *Geophysical Research Letters*, *41*(20), 7306–7313.
URL https://agupubs.onlinelibrary.wiley.com/doi/abs/10.1002/2014GL061413

Ragon, C., Lembo, V., Lucarini, V., Vérard, C., Kasparian, J., & Brunetti, M. (2022). Robustness of competing climatic states. *Journal of Climate*, *35*(9), 2769 – 2784.
URL https://journals.ametsoc.org/view/journals/clim/35/9/JCLI-D-21-0148.1.xml

Raup, D. M. (1979). Size of the permo-triassic bottleneck and its evolutionary implications. *Science*, *206*(4415), 217–218.
URL https://www.science.org/doi/abs/10.1126/science.206.4415.217

Reichow, M. K., Pringle, M., Al'Mukhamedov, A., Allen, M., Andreichev, V., Buslov, M., Davies, C., Fedoseev, G., Fitton, J., Inger, S., Medvedev, A., Mitchell, C., Puchkov, V., Safonova, I., Scott, R., & Saunders, A. (2009). The timing and extent of the eruption of the siberian traps large igneous province: Implications for the end-permian environmental crisis. *Earth and Planetary Science Letters*, *277*(1), 9–20.
URL https://www.sciencedirect.com/science/article/pii/S0012821X08006389

Renne, P. R., Black, M. T., Zichao, Z., Richards, M. A., & Basu, A. R. (1995). Synchrony and causal relations between permian-triassic boundary crises and siberian flood volcanism. *Science*, *269*(5229), 1413–1416.
URL https://www.science.org/doi/abs/10.1126/science.269.5229.1413

Retallack, G. J., Sheldon, N. D., Carr, P. F., Fanning, M., Thompson, C. A., Williams, M. L., Jones, B. G., & Hutton, A. (2011). Multiple early triassic greenhouse crises impeded recovery from late permian mass extinction. *Palaeogeography, Palaeoclimatology, Palaeoecology*, *308*(1), 233–251.
URL https://www.sciencedirect.com/science/article/pii/S0031018210005924

Romano, C., Goudemand, N., Vennemann, T. W., Ware, D., Schneebeli-Hermann, E., Hochuli, P. A., Brühwiler, T., Brinkmann, W., & Bucher, H. (2013). Climatic and biotic upheavals following the end-permian mass extinction. *Nature Geoscience*, *6*(1), 57–60.

Royer, D. L., Berner, R. A., Montañez, I. P., Tabor, N. J., Beerling, D. J., et al. (2004). Co˜ 2 as a primary driver of phanerozoic climate. *GSA today*, *14*(3), 4–10.

Salzmann, U., Haywood, A. M., Lunt, D. J., Valdes, P. J., & Hill, D. J. (2008). A new global biome reconstruction and data-model comparison for the middle pliocene. *Global Ecology and Biogeography*, *17*(3), 432–447.
URL https://onlinelibrary.wiley.com/doi/abs/10.1111/j.1466-8238.2008.00381.x

Schneebeli-Hermann, E. (2020). Regime shifts in an early triassic subtropical ecosystem. *Frontiers in Earth Science*, *8*.
URL https://www.frontiersin.org/articles/10.3389/feart.2020.588696

Sellwood, B. W., & Valdes, P. J. (2008). Jurassic climates. *Proceedings of the Geologists' Association*, *119*(1), 5–17.
URL https://www.sciencedirect.com/science/article/pii/S0016787859800687

Stanley, S. M. (2016). Estimates of the magnitudes of major marine mass extinctions in earth history. *Proceedings of the National Academy of Sciences*, *113*(42), E6325–E6334.
URL https://www.pnas.org/doi/abs/10.1073/pnas.1613094113

Sun, Y., Joachimski, M. M., Wignall, P. B., Yan, C., Chen, Y., Jiang, H., Wang, L., & Lai, X. (2012). Lethally hot temperatures during the early triassic greenhouse. *Science*, *338*(6105), 366–370.
URL https://www.science.org/doi/abs/10.1126/science.1224126

Trenberth, K. E. (2020). Understanding climate change through earth's energy flows. *Journal of the Royal Society of New Zealand*, *50*(2), 331–347.
URL https://doi.org/10.1080/03036758.2020.1741404

Widmann, P., Bucher, H., Leu, M., Vennemann, T., Bagherpour, B., Schneebeli-Hermann, E., Goudemand, N., & Schaltegger, U. (2020). Dynamics of the largest carbon isotope excursion during the early triassic biotic recovery. *Frontiers in Earth Science*, *8*.
URL https://www.frontiersin.org/articles/10.3389/feart.2020.00196

Ziegler, A. M. (1990). Phytogeographic patterns and continental configurations during the permian period. *Geological Society, London, Memoirs*, *12*(1), 363–379.
URL https://www.lyellcollection.org/doi/abs/10.1144/GSL.MEM.1990.012.01.35

---

## Author Comment (AC3)

Manuscript: egusphere-2023-1808

**Title:** Alternative climatic steady states for the Permian-Triassic paleogeography

**Reviewer #3:**

The presented manuscript describes how up to three different climatic states exist under the same forcing in a climate model of the Permian-Triassic paleogeography. Multistability is further shown to exist also when vegetation and carbon cycle feedbacks are activated and the existence of multiple stable states is proposed as an explanation for the observed high climate variability of the PTB, as well as discrepancies between numerical simulations and geological data. The authors use MITgcm, a coupled general circulation model with a coarse resolution that properly resolves the ocean circulation, but only has a relatively simple representation of atmospheric dynamics (e.g. only 5 vertical layers).

The manuscript is generally well-structured and easy to follow. The language and grammar probably need a revision. The finding of several (non-snowball Earth) stable climate equilibria at the time of the PTB is interesting and, to my knowledge, has not been shown before.

We thank the reviewer for this overall positive statement on the originality of our work, and we are willing to improve the language of our manuscript and to respond to the following criticisms.

However, I have a major concern about the validity of the findings, coming from some of the presented simulation results:

i.) The "hot" state has a global mean temperature of 30.9 °C at a $CO_2$ concentration of 320 ppm and a solar forcing that is $\sim 2\%$ weaker than the present-day forcing. To put this in other words, even though the $CO_2$ concentration is 20-25% lower than the modern values and the sun is 2% weaker than today (equivalent to roughly another halving of the $CO_2$ concentration), the global climate of the hot state is simulated to be approximately 16 °C warmer than the present-day climate. I am aware that a different continental distribution can lead to very different global mean temperatures, but this contrast to the present-day climate is so extreme that I am really wondering where this is coming from? An explanation for this extreme state is not given in the manuscript. In general, all three states ("cold", "warm" and "hot") have pretty high global mean temperatures. Is it because the land surface has a comparably low albedo value? Does the continental distribution lead to a much smaller cloud coverage? Or is there a very strong water vapour feedback in the MITgcm atmosphere component? A closer investigation of this aspect would, in my opinion, not just be interesting, but is actually crucial, as otherwise the validity of the model results is very questionable.

The reviewer is right saying that more details are needed on the physical mechanisms to explain the origin of the three different steady states and epecially the hot one. The mean global air surface temperature of the cold state at 320 ppm, equal to 17.2 °C, is already larger than the present-day value. We have described in Section 3.1 many different quantities which characterise the ocean and atmosphere dynamics. For example, in the cold state the ocean circulation is completely different from the present-day one, with a single anti-clockwise overturning cell (Fig. 6f) and the absence of circumpolar currents (Fig. 10c) because of the presence of the Pangea continent. It is clear that the boundary conditions gives rise to crucial differences with the present-day climate, affecting the heat transport (Fig. 5) and the asymmetrical formation of sea ice in the two hemispheres.

In addition, alternative climatic states can be realised under the same boundary conditions and the same forcing. This is due to the fact that there are many nonlinear mechanisms acting between the climatic spheres (atmosphere, cryosphere, hydrosphere..) on a given time scale, and they can balance in different ways under the same forcing, as is well known in climate modelling (within the whole hierarchy of models, starting from the energy balance models towards the complex general circulation models). For example, multiple steady states have been obtained using different MITgcm setups in coupled-aquaplanet configurations (Ferreira et al., 2011; Brunetti et al., 2019; Ragon et al., 2022; Brunetti & Ragon, 2023; Zhu & Rose, 2023) or other models (Popp et al., 2016; Lucarini & Bódai, 2020), includind high compexity models as IPCC-class models (Peltier & Vettoretti, 2014; Jackson & Wood, 2018; Malmierca-Vallet et al., 2023).

In a coupled aquaplanet (where the full nonlinear feedbacks between ocean, atmosphere and sea ice are taken into account over a millennial time scale), we investigated the role of cloud albedo (Brunetti et al., 2019) and indeed showed that the presence of the hot state depends on cloud reflection properties and on the amount of solar radiation that is allowed to enter at high latitudes (see also Ragon et al. (2022)).

Following the suggestion of the reviewer, we have analysed the cloud cover in the three steady states. It is larger in the hot state compared to the cold state in polar regions and on land (see Fig. 1a in this

response). The planetary albedo is 30% in the cold state, 29% in the warm state and 27% in the hot state, which are ≤ 30%, the present-day estimation (Goode et al., 2001). Thus, the energy absorbed into the atmosphere is generally larger than for present-day climate, and is larger in the hot state than in the cold one. At the same time, the atmospheric transmissivity of long-wave radiation is smaller in the hot state (0.50) than in the cold one (0.57), meaning that more thermal radiation is trapped within the atmosphere in the hot state. The combined effects of cloud feedback lead to more radiation to enter and stay into the atmosphere, the cloud feedback becoming dominant in the hot state, as also observed in the coupled-aquaplanet configuration (Brunetti et al., 2019). We suggest to add the analysis of the cloud cover in the three states in Section 3.1 of the manuscript.

[Figure]

Figure 1: Cloud cover in (a) hot-cold; (b) warm-cold; (c) cold state.

ii.) The plot of global mean temperatures in Fig. 11 and the values in Tab. 5 highlight another extreme (maybe even unrealistic?) aspect of the simulated model results: the climate sensitivity seems to be extremely high. The slopes of the lines in Fig. 11 indicate that the climate sensitivity of this setup is 15-18 °C of warming per doubling of $CO_2$. An increase of just 8 ppm in the "warm" state led to a temperature increase of 1.45 °C (Tab. 5). If the modern climate would be anywhere near this sensitivity, humanity would be doomed already (we added ∼ 140 ppm carbon to experience a comparable warming). The most recent IPCC estimate of modern climate sensitivity is 2.5-4 °C (likely range). Again, a different continental setup might explain some of the differences to the modern state, but the climate sensitivity in this study is so extreme that it requires a convincing explanation. Otherwise the model results cannot be viewed as reliable.

We agree that the equilibrium climate sensitivity of the three climatic states (from the slopes of the lines in Fig. 11) is quite large in comparison with the IPCC estimate obtained with CMIP-type models for present day. However, we need to consider that:

1. There is a spread in model results for the present-day climate, as summarised in Knutti et al. (2017), depending on model complexity and considered time scale.

2. The simulations of the present-day climate are tuned using many available observations. This is not possible of course for the Permian-Triassic period.

3. The atmospheric module is based on SPEEDY, the Simplified Parametrizations primitivE-Equation DYnamics described in Molteni (2003) (see also the Appendix to that article on the web page of SPEEDY), with 5 vertical layers. It is important to consider that in the parameterization scheme for the longwave radiation, the infrared spectrum is partitioned into four regions: *i)* the 'infrared window' between 8.5 and 11 $\mu$m; *ii)* the band of strong absorption by $CO_2$ around 15 $\mu$m; *iii)* the aggregation of regions with weak/moderate absorption by water vapour; *iv)* the aggregation of regions with strong absorption by water vapour. Thus, the absorption of $CO_2$ is limited by the fact that only the largest absorption band is included in the infrared spectrum. This implies that the atmospheric $CO_2$ content is probably underestimated in our simulations, and that the range of $CO_2$ for the steady states can change when considering the full absorption bands. This also affects the equilibrium climate sensitivity of the MITgcm. We can include this more detailed description of the infrared spectrum in SPEEDY in Section 2.1 in the manuscript. The advantage of SPEEDY is that it requires one order of magnitude less CPU time than a state-of-the-art GCM at the same horizontal resolution, and is therefore suitable for studies on millennial time scale.

4. Despite this simplified atmospheric module, we have reproduced the present-day climate with our MITgcm setup (see Brunetti & Vérard (2018) and additional comments below). Thus, we are confident that our model results are reliable, the main feedback mechanisms being properly described at

main order (within the limitations of low-resolution general circulation models (GCMs) and simplified parameterizations discussed above, typical of other Earth Models of Intermediate Complexity (EMICs)).

5. The simplified atmospheric module and the low spatial resolution likely affect the forcing range of stability, as we said above, but we expect that they have a small effect on the overall structure of the bifurcation diagram. The reason is that the presence of multiple steady states depends on the number of feedbacks included in the simulation on a given time scale and how they can balance between each other.

6. It is important to repeat our simulations with other models with different complexity (EMICs, CMIP-type GCMs, low-resolution GCMs, ...) in order to investigate the robustness of the alternative climatic states, as we state in the Conclusions (line 299). Other modellers of past climates have already shown their interest in the multistability framework that we propose in the present manuscript, and it is important to publish results obtained with different models. We are also involved in the Swiss National Science Foundation (SNSF) Sinergia Project n. 213539 where we plan to compare FOAM, PlaSim/cGENIE and MITgcm in future publications (at different time slices including present day). Some of the authors are also involved in TIPMIP, the Tipping Point Modelling Intercomparison Project.

iii.) I find it quite surprising that the deserts are smallest in the "hot" state. The authors claim that this is because there is more precipitation in that state. However, the annual mean surface air temperatures are 40-50 °C (daily temperature extremes should then be around 70-80 °C) in tropical and subtropical regions, where their model is simulating a vegetation cover of "forbland and dry shrubland". I am no expert in vegetation cover, but could any plant survive temperatures of >70 °C, even if it is just for a few hours a day? Additionally, even though there might be more precipitation in the hot state, this would probably be more localized in individual extreme events and not fall evenly. I see a potential discrepancy here, because the vegetation model is only fed with long-term monthly mean values, which don't capture this variability.

The annual mean surface air temperature (SAT) in the hot state (after convergence with BIOME4) is 32.4 °C (see Table 4), and not 40-50 °C. Seasonal precipitation and SAT for the hot and cold states are shown in Figs. 2, 3 in this response, respectively, while min/max/mean values of monthly averaged SAT and precipitation are shown in Fig. 4. Regions of null precipitation are the same along all the year in the cold state (Fig. 2, bottom panels), within 20 and 40 degLat, with mean SAT of the order of 40 °C (Fig. 3, bottom panels). These regions correspond to desert in the cold state (Fig. 12 in the manuscript).

Regions of null precipitation have a much reduced extent depending on the season in the hot state (Fig. 2 in this response, upper panels), despite reaching higher temperature than in the cold state, the maximum being of the order of 60 °C (Fig. 4). This is why desert has less extent in the hot state, keeping also in mind that SAT is not the only driver of desert conditions, soil temperature and availability of water being crucial as well for the plant development (Wahid et al., 2007; Hatfield & Prueger, 2015).

The biome denoted as 'Forbland and dry shrubland' in Harrison & Prentice (2003) (the same classification that is used in Fig. 12 in our manuscript) contains tropical and temperate xerophytic shrubland (that is, shrubs adapted to dry periods), but also tropical and temperate forbland. Thus, this kind of biome is sufficiently general to appear in the hot state.

iv.) It is also quite unusual that the meridional overturning circulation (MOC) of the ocean is much stronger in the hot state, especially since there is a weakening of the MOC, when going from the cold to the warm state. Furthermore, in the hot state there seems to be a cell near the equator where the water transport is out of the bounds of the colorbar, i.e. has a transport of >100 Sv, which is extremely high and - to my knowledge - unrealistic for a supposed equilibrium state. Where is the energy that drives a constant massive ocean overturning coming from?

We have compared our results with those in Hülse et al. (2021) (see in particular Fig. 3 in that paper and lines 189-191 in our manuscript). They consider the Permian-Triassic paleogeography and different values of atmospheric $CO_2$ content. By increasing $CO_2$, the ocean overturning circulation changes structure: a single counter-clockwise overturning cell at low $CO_2$ (as in our cold state), which increases in intensity as a clockwise cell appears first at the North polar region (warm state) and then in the whole Northern hemisphere at high $CO_2$ (as in our hot state). This shows that the overall structure of the overturning cells in our simulations corresponds well to that in Hülse et al. (2021).

[Figure]

Figure 2: Seasonal (DJF: a,c; JJA: b,d) precipitation in hot (a-b) and cold (c-d) states when convergence with BIOME4 is attained.

[Figure]

Figure 3: Seasonal (DJF: a,c; JJA: b,d) surface air temperature in hot (a-b) and cold (c-d) states when convergence with BIOME4 is attained.

Note that the intensity of local maximum of the overturning cell around 50S in the three attractors increases going from cold (19 Sv) to warm (25 Sv) and hot (33 Sv) (see Fig. 6 in the manuscript, bottom panels), this trend being in agreement with Hülse et al. (2021).

In order to investigate the reason why the overturning cell near the Equator is so intense in the hot state, we have performed a detailed study of the overturning circulation by first analysing the contribution of Tethys and Panthalassa. It turns out that the intense overturning cell is due to the circulation in

[Figure]

[Figure]

Figure 4: Monthly mean, minimum and maximum values of surface air temperature (left) and precipitation (right) for hot and cold states.

[Figure]

Figure 5: Residual-mean circulation in Panthalassa for the hot (a), warm (b) and cold (c) states.

Panthalassa because of the presence of equatorial oceanic ridges which can be seen in Fig. 1 in our manuscript on the west of Pangea. The overturning streamfunction in Fig. 6 (bottom panels) of the manuscript is calculated using the mean meridional velocity component. However, there is a turbulent component that is particularly strong near the west coast of Pangea at depth. Our preliminary results show that the residual-mean circulation, given by the sum of the mean part and the turbulent circulation (Danabasoglu et al., 1994; Ferreira et al., 2011) is much less intense (the intensity of the local maximum is of the order of 90 Sv), as can be seen in Fig. 5.

At this point, I am having a hard time believing the outcome of the simulation results, especially with respect to the high climate sensitivity and the generally unusual characteristics of the "hot" state. When seeing these results, my first guess would be that the whole "hot" state is a numerical artefact and that the atmosphere component of MITgcm is not doing a decent job here in general. In order to oppose my concern, the authors could present results of a reference simulation of 1850-today, to show that this version of MITgcm (with the necessary adaptions to the modern state) is able to get the historical warming roughly right (or one pre-industrial simulation and one with a more recent $CO_2$ concentration at 400 ppm, if that is easier).

A pre-industrial simulation obtained with MITgcm in a coupled atmosphere-ocean-sea ice-land configuration similar to the one used here has been published in Brunetti & Vérard (2018) for $CO_2 = 326$ ppm and $S_0 = 342$ W/m$^2$ (cf. in particular the setup denoted as `Run4` in that paper). The simulation reproduces reasonably well the pre-industrial conditions, in particular the mean surface air temperature, the sea ice extent in the Arctic and Antarctic regions, the structure and maximal intensity of the overturning circulation, as expected by a low-resolution GCM (2.8°, 5 levels in the atmosphere, 15 in the ocean) with simplified atmospheric parametrizations. In addition, the TOA imbalance is quite low (-0.55 W/m$^2$), as well as the surface energy imbalance (0.04 W/m$^2$), assuring a small temperature drift of 0.009 K/century (Table 3 in Brunetti & Vérard (2018)).

We have not a simulation at 400 ppm (the simulations in Brunetti & Vérard (2018) were run with version MITgcm_c65q, while for changing the $CO_2$ content a more recent version is needed). Preliminary results with MITgcm_c67f show that the climate sensitivity may be large (without any particular effort of tuning to the present-day conditions). We plan to perform a detailed study of the climate sensitivity

and the complete analysis of the steady states for the present-day configuration in the context of the SNSF Sinergia Project that we mentioned above and just started in October 2023. However, this study requires time and is beyond the scope of the present manuscript.

Additionally, I really need to see a discussion of how the extreme and unusual results of the hot state can be explained physically. As the existence of the "hot" state at such low CO2 concentrations is almost impossible in my eyes and given the other strange features mentioned above, I highly doubt the reliability of the presented results. Since the whole point of this manuscript is based on the existence of multistability, I have no other option than to suggest a rejection of the manuscript, unless the authors provide an elaborate and convincing explanation of why these extreme results are realistic.

We believe that our responses to all the previous points provide physical explanations for the existence of the hot state, in particular the analysis of the cloud feedbacks (point (i)), of the seasonal and monthly averages of surface air temperature and precipitation (point (iii)), and of the overturning circulation (point (iv)). All these detailed analyses improve the quality of our paper and we are grateful to the reviewer for this.

Moreover, we have explored the existence of multiple steady states in a coupled aquaplanet using the MITgcm in previous studies. We have shown in particular that the hot state depends on the amount of solar energy allowed to enter in the polar regions, thus on the cloud description (Brunetti et al., 2019); from a detailed analysis performed in Ragon et al. (2022), the hot state maximises not only the global temperature, but also the Material Entropy Production (MEP) due to the hydrological cycle (which represents the largest contribution to the total MEP), the total precipitation and the peak intensity in the water-mass transport. Other steady states maximise other quantities, each state being the result of a different balance between nonlinear feedbacks.

Finally, geological data suggest that climatic oscillations occurred in the Early Triassic, with temperature fluctuations of the order of ten degrees (see also a comment below, p. 7), from 'hot' to 'very hot' conditions. We believe that the multistability framework is particularly relevant for explaining such climatic oscillations.

Of course, the same procedure used in our manuscript should be repeated with other models of different complexity. However, the construction of the bifurcation diagram takes time. We believe that it is important to open the way, and show that it is feasible by providing to the scientific community a first detailed study (with all its limitations that we think we have honestly described) to start with.

**Specific comments**

- The language/grammar of the manuscript needs a revision. Very often I have the feeling that a "the" or similar article is missing (as an example in line 184 the sentence should be "In the hot state,...", right?).

We will correct this and similar errors. We shall give the final version of the manuscript to English speaking colleagues for a check.

- The whole discussion neglects the fact that there should be another stable climate: the snowball Earth. This should be mentioned at least once at some point.

We agree with the reviewer, and we will mention this possibility. However, we restricted our analysis to the steady states of relevance to the Early Triassic. We confirm that at least a colder climate exists: at the lower edge of the stable branches of both hot and cold states the system is attracted towards a colder state, as shown in Fig. B1 of the manuscript. A waterbelt state is present where the ice extends to $\sim 30°$ latitude and the global mean surface air temperature is approx. $-10\ °C$ (see Fig. 6 in this response). However, we have not investigated this state further since simulations require long CPU time while geological data exclude for the presence of snowball or waterbelt states in the Early Triassic (Sun et al., 2012; Romano et al., 2013; Goudemand et al., 2019; Widmann et al., 2020). Similarly, we did not search for steady states at global SAT below $-10\ °C$, but there is no reason to expect that no snowball exists.

- I would appreciate some more introduction as to why the Permian-Triassic Boundary is an important/interesting period to study? Right now the introduction is only about multistability and tipping points.

We propose to include the following paragraph in the Introduction (at line 40) to describe the climatic oscillations in the aftermath of the Permian-Triassic Boundary mass extinction and to provide the general context for our numerical simulations.

*We are interested in the climatic oscillations observed at the Smithian-Spathian boundary in the Early Triassic, just after the Permian-Triassic boundary (PTB) mass extinction ($\sim$252 Ma), the most severe*

[Figure]

Figure 6: Sea-surface temperature and sea-ice extent in the waterbelt state at global mean SAT $\sim -10\,°$C.

*of the Phanerozoic Raup (1979); MacLeod (2014); Stanley (2016). As a consequence of the volcanic activity of the Siberian Large Igneous Province Campbell et al. (1992); Renne et al. (1995); Reichow et al. (2009), the global carbon cycle entered a perturbed state which persisted for nearly 5.4 Myr in the Early Triassic, until a new equilibrium state was reached in the Anisian Sun et al. (2012); Romano et al. (2013); Goudemand et al. (2019); Leu et al. (2019); Widmann et al. (2020). The observed fluctuations in the carbon isotope record Payne et al. (2004); Galfetti et al. (2007); Retallack et al. (2011) with successive diversification-extinction cycles of the nekton Orchard (2007); Brühwiler et al. (2010); Leu et al. (2019) and ecological crises of terrestrial plants Hermann et al. (2011); Schneebeli-Hermann (2020), are all indicative of the climatic changes which occurred in the Early Triassic, with variations in the global temperature of the order of 7-8 °C from the thermal maximum in the late Smithian to cold climates in early Spathian times Widmann et al. (2020).*

*However, despite the great effort of the scientific community to reconstruct such climatic oscillations in the aftermath of the PTB, large uncertainties remain in timing and causal relationships, implying that numerical modelling needs to consider a wide range of initial and boundary conditions for the simulation of such geological interval. In this context, ...* [continuing at line 44]

- The model has no sea-ice dynamics, which would strongly impact individual climate states that have some sea ice. Given this shortcoming, also the small range in which the "warm" state exists is questionable.

We agree that the number of steady states and the extension of their stability region may depend on the configuration and the model setup. Indeed, feedback mechanisms acting on the same time scale could affect the balance between different processes, and thus the number of steady states. A sentence on this aspect can be added in the Conclusions of the manuscript, mentioning the role of sea ice dynamics, and also ocean dynamics, and relevant references (at line 300):

*.... boundary conditions. In particular, including sea ice dynamics or different numerical implementations of thermodynamic sea ice Lewis et al. (2007); Voigt & Abbot (2012); Hörner & Voigt (2023), as well as considering a mixed-layer ocean or a fully dynamical one Poulsen et al. (2001); Pohl et al. (2014), may change the number of steady states, and reveal the source of possible biases.*

This is also the reason why in the Conclusions (last paragraph, line 299) we call for the need of comparing different climate models. We can also specify that we use a *thermodynamic* sea ice in the abstract and the Conclusions (second paragraph, line 278) when we mention the MITgcm configuration.

- the choice of colors in the upper panel of Fig. 2, using darker colors for a higher albedo, seems odd. I would reverse the colorbar

We agree with this suggestion and we will reverse the colors.

- line 143: averaged over which time scale?

The average is computed over the last 100 yr of that part of the simulation (step 1 of the air-sea carbon exchange procedure).

- a short description of how the carbon cycle model works would be very helpful. Right now, I cannot judge whether this model is sufficient to provide a proper representation of the carbon cycle during the PTB

The procedure for including air-sea carbon exchanges is described in section 2.3 and it consists of two steps: 1) we estimate the distributions of oceanic tracers (DIC, dissolved inorganic phosphorous, alkalinity, phosphate and oxygen), which correspond to given values of atmospheric $CO_2$ content on the stable branch of the attractors. 2) These distributions are then used when the air-sea carbon exchanges are allowed. By construction, air-sea carbon exchanges vary the atmospheric $CO_2$ content of the order of 10 ppm (see Table 5). The advantage of including the option of $CO_2$ exchanges is to obtain the total (ocean + atmosphere) carbon content for each state, as reported in Table 5.

- lines 239-248: how do the plants survive these extreme temperatures in the warm state? Also, there is more carbon stored in vegetation in the hot state. This also means there is a lot of fuel for fires, which should occur very often given the extremely high temperatures. Is this self-consistent?

For the first question, see answer to point (iii).

We started to investigate the question of fires and possible geological signatures in collaboration with biochemists at the University of Lausanne. In order to have fires burning a large amount of biomass over long time scales (millennial), a season of vegetation development should be followed by a dry and hot one. This can indeed be the case in the tropical and subtropical regions of the hot state. However, the dominant biome in such regions is 'forbland and dry shurland', which has not a high biomass density (see Table D1 in our manuscript). Thus, the overall biomass content in the hot state can only slightly vary over millennial time scales.

- Figure captions could include more information to make the figures a bit more self-evident (e.g. Fig. 7-9)

We shall add such information.

- adding some contour lines in e.g. Fig. 3-4 would really help for readability

We thank the reviewer for this suggestion. Figures 3, 4, and also Fig. 9 (salinity distribution) in the manuscript can be modified as shown in Fig. 7, 8, 9 in this response, respectively.

[Figure]

Figure 7: Sea-surface temperature and sea-ice thickness for (a) hot, (b) warm, and (c) cold states. White area corresponds to land.

- The first paragraph of the conclusion is rather a repetition of the introduction

Even if the content is similar, it is differently formulated. We think it is a useful summary of the main points and we would like to keep as it is.

- I don't understand the way that uncertainty in numbers is represented. For example in Tab.4: in the initial state of the hot state, the second iteration of the warm state and the fourth iteration of the cold state the uncertainty of SAT is much larger (9 instead of 1) than in the other states. Do these three cases actually have an uncertainty of $0.9°C$ and not $9°C$? Otherwise the results seem to suggest some kind of instability. Do I understand those numbers correctly?

[Figure]

Figure 8: Near-surface air temperature for (a) hot, (b) warm, and (c) cold states.

[Figure]

Figure 9: Sea-surface salinity of (a) hot, (b) warm, and (c) cold states. White area corresponds to land.

The number in parenthesis is the uncertainty associated to the last significant digit. For example, in the initial hot state, 30.92 (9) °C corresponds to (30.92±0.09) °C. This is a compact way to write uncertainties which is commonly used in physics and climate literature. While we expect that this compact form helps legibility, we are open to change to the extended form if required to do so.

- The river map in Fig. A1 could simply be added to Fig. 1. The Appendix A is then obsolete, as the content is also already mentioned in the model description section.
We agree with this suggestion.

- What about other greenhouse gases? Are they held fixed at some values? Which values were used?
The longwave radiation scheme in the atmospheric module (SPEEDY, Molteni (2003)) uses four spectral bands, as described in this response (point *ii*.3). The absorption by other greenhouse gases as methane is not included. This information can be added in section 2.1 when we mention the SPEEDY module.

**Some technical corrections:**

- labelling subfigures in Fig. 2 with (a) and (b)
We thank the reviewer for pointing this out.

- Fig. 10: It says that the line thickness varies with horizontal velocity, but it is not. Maybe make the thickness more sensitive to velocity or just drop the velocity scaling.
We agree that the variation in line thickness is not very visible. We can drop the velocity scaling.

**References**

Brunetti, M., Kasparian, J., & Vérard, C. (2019). Co-existing climate attractors in a coupled aquaplanet. *Climate Dynamics*, *53*(9-10), 6293–6308.
URL http://link.springer.com/10.1007/s00382-019-04926-7

Brunetti, M., & Ragon, C. (2023). Attractors and bifurcation diagrams in complex climate models. *Phys. Rev. E*, *107*, 054214.
URL https://link.aps.org/doi/10.1103/PhysRevE.107.054214

Brunetti, M., & Vérard, C. (2018). How to reduce long-term drift in present-day and deep-time simulations? *Climate dynamics*, *50*(11-12), 4425–4436.

Brühwiler, T., Bucher, H., Brayard, A., & Goudemand, N. (2010). High-resolution biochronology and diversity dynamics of the early triassic ammonoid recovery: The smithian faunas of the northern indian margin. *Palaeogeography, Palaeoclimatology, Palaeoecology*, *297*(2), 491–501.
URL https://www.sciencedirect.com/science/article/pii/S0031018210005274

Campbell, I. H., Czamanske, G. K., Fedorenko, V. A., Hill, R. I., & Stepanov, V. (1992). Synchronism of the siberian traps and the permian-triassic boundary. *Science*, *258*(5089), 1760–1763.
URL https://www.science.org/doi/abs/10.1126/science.258.5089.1760

Danabasoglu, G., McWilliams, J. C., & Gent, P. R. (1994). The role of mesoscale tracer transports in the global ocean circulation. *Science*, *264*(5162), 1123–1126.

Ferreira, D., Marshall, J., & Rose, B. (2011). Climate determinism revisited: Multiple equilibria in a complex climate model. *Journal of Climate*, *24*(4), 992 – 1012.
URL https://journals.ametsoc.org/view/journals/clim/24/4/2010jcli3580.1.xml

Galfetti, T., Bucher, H., Ovtcharova, M., Schaltegger, U., Brayard, A., Brühwiler, T., Goudemand, N., Weissert, H., Hochuli, P. A., Cordey, F., & Guodun, K. (2007). Timing of the early triassic carbon cycle perturbations inferred from new u–pb ages and ammonoid biochronozones. *Earth and Planetary Science Letters*, *258*(3), 593–604.
URL https://www.sciencedirect.com/science/article/pii/S0012821X0700252X

Goode, P. R., Qiu, J., Yurchyshyn, V., Hickey, J., Chu, M.-C., Kolbe, E., Brown, C. T., & Koonin, S. E. (2001). Earthshine observations of the earth's reflectance. *Geophysical Research Letters*, *28*(9), 1671–1674.
URL https://agupubs.onlinelibrary.wiley.com/doi/abs/10.1029/2000GL012580

Goudemand, N., Romano, C., Leu, M., Bucher, H., Trotter, J. A., & Williams, I. S. (2019). Dynamic interplay between climate and marine biodiversity upheavals during the early triassic smithian -spathian biotic crisis. *Earth-Science Reviews*, *195*, 169–178.
URL https://www.sciencedirect.com/science/article/pii/S0012825218302563

Harrison, S. P., & Prentice, C. I. (2003). Climate and co2 controls on global vegetation distribution at the last glacial maximum: analysis based on palaeovegetation data, biome modelling and palaeoclimate simulations. *Global Change Biology*, *9*(7), 983–1004.

Hatfield, J. L., & Prueger, J. H. (2015). Temperature extremes: Effect on plant growth and development. *Weather and Climate Extremes*, *10*, 4–10. USDA Research and Programs on Extreme Events.
URL https://www.sciencedirect.com/science/article/pii/S2212094715300116

Hermann, E., Hochuli, P. A., Bucher, H., Brühwiler, T., Hautmann, M., Ware, D., & Roohi, G. (2011). Terrestrial ecosystems on north gondwana following the end-permian mass extinction. *Gondwana Research*, *20*(2), 630–637.
URL https://www.sciencedirect.com/science/article/pii/S1342937X1100030X

Hörner, J., & Voigt, A. (2023). Sea-ice thermodynamics can determine waterbelt scenarios for snowball earth. *EGUsphere*, *2023*, 1–13.
URL https://egusphere.copernicus.org/preprints/2023/egusphere-2023-2073/

Hülse, D., Lau, K. V., van de Velde, S. J., Arndt, S., Meyer, K. M., & Ridgwell, A. (2021). End-permian marine extinction due to temperature-driven nutrient recycling and euxinia. *Nature Geoscience*, *14*(11), 862–867.

Jackson, L. C., & Wood, R. A. (2018). Hysteresis and resilience of the amoc in an eddy-permitting gcm. *Geophysical Research Letters*, *45*(16), 8547–8556.
URL https://agupubs.onlinelibrary.wiley.com/doi/abs/10.1029/2018GL078104

Knutti, R., Rugenstein, M. A., & Hegerl, G. C. (2017). Beyond equilibrium climate sensitivity. *Nature Geoscience*, *10*(10), 727–736.

Leu, M., Bucher, H., & Goudemand, N. (2019). Clade-dependent size response of conodonts to environmental changes during the late smithian extinction. *Earth-Science Reviews*, *195*, 52–67.
URL https://www.sciencedirect.com/science/article/pii/S0012825218302216

Lewis, J. P., Weaver, A. J., & Eby, M. (2007). Snowball versus slushball earth: Dynamic versus nondynamic sea ice? *Journal of Geophysical Research: Oceans*, *112*(C11).
URL https://agupubs.onlinelibrary.wiley.com/doi/abs/10.1029/2006JC004037

Lucarini, V., & Bódai, T. (2020). Global stability properties of the climate: Melancholia states, invariant measures, and phase transitions. *Nonlinearity*, *33*(9), R59.

MacLeod, N. (2014). The geological extinction record: History, data, biases, and testing. In *Volcanism, Impacts, and Mass Extinctions: Causes and Effects*. Geological Society of America.
URL https://doi.org/10.1130/2014.2505(01)

Malmierca-Vallet, I., Sime, L. C., & the D-O community members (2023). Dansgaard–oeschger events in climate models: review and baseline marine isotope stage 3 (mis3) protocol. *Climate of the Past*, *19*(5), 915–942.
URL https://cp.copernicus.org/articles/19/915/2023/

Molteni, F. (2003). Atmospheric simulations using a GCM with simplified physical parametrizations. I: model climatology and variability in multi-decadal experiments. *Climate Dynamics*, *20*(2), 175–191.
URL http://link.springer.com/10.1007/s00382-002-0268-2

Orchard, M. J. (2007). Conodont diversity and evolution through the latest permian and early triassic upheavals. *Palaeogeography, Palaeoclimatology, Palaeoecology*, *252*(1), 93–117.
URL https://www.sciencedirect.com/science/article/pii/S0031018207001113

Payne, J. L., Lehrmann, D. J., Wei, J., Orchard, M. J., Schrag, D. P., & Knoll, A. H. (2004). Large perturbations of the carbon cycle during recovery from the end-permian extinction. *Science*, *305*(5683), 506–509.
URL https://www.science.org/doi/abs/10.1126/science.1097023

Peltier, W. R., & Vettoretti, G. (2014). Dansgaard-oeschger oscillations predicted in a comprehensive model of glacial climate: A "kicked" salt oscillator in the atlantic. *Geophysical Research Letters*, *41*(20), 7306–7313.
URL https://agupubs.onlinelibrary.wiley.com/doi/abs/10.1002/2014GL061413

Pohl, A., Donnadieu, Y., Le Hir, G., Buoncristiani, J.-F., & Vennin, E. (2014). Effect of the ordovician paleogeography on the (in)stability of the climate. *Climate of the Past*, *10*(6), 2053–2066.
URL https://cp.copernicus.org/articles/10/2053/2014/

Popp, M., Schmidt, H., & Marotzke, J. (2016). Transition to a Moist Greenhouse with CO2 and solar forcing. *Nature Communication 7*, *10627*.

Poulsen, C. J., Pierrehumbert, R. T., & Jacob, R. L. (2001). Impact of ocean dynamics on the simulation of the neoproterozoic "snowball earth". *Geophysical Research Letters*, *28*(8), 1575–1578.
URL https://agupubs.onlinelibrary.wiley.com/doi/abs/10.1029/2000GL012058

Ragon, C., Lembo, V., Lucarini, V., Vérard, C., Kasparian, J., & Brunetti, M. (2022). Robustness of competing climatic states. *Journal of Climate*, *35*(9), 2769 – 2784.
URL https://journals.ametsoc.org/view/journals/clim/35/9/JCLI-D-21-0148.1.xml

Raup, D. M. (1979). Size of the permo-triassic bottleneck and its evolutionary implications. *Science*, *206*(4415), 217–218.
URL https://www.science.org/doi/abs/10.1126/science.206.4415.217

Reichow, M. K., Pringle, M., Al'Mukhamedov, A., Allen, M., Andreichev, V., Buslov, M., Davies, C., Fedoseev, G., Fitton, J., Inger, S., Medvedev, A., Mitchell, C., Puchkov, V., Safonova, I., Scott, R., & Saunders, A. (2009). The timing and extent of the eruption of the siberian traps large igneous province: Implications for the end-permian environmental crisis. *Earth and Planetary Science Letters*, *277*(1), 9–20.
URL https://www.sciencedirect.com/science/article/pii/S0012821X08006389

Renne, P. R., Black, M. T., Zichao, Z., Richards, M. A., & Basu, A. R. (1995). Synchrony and causal relations between permian-triassic boundary crises and siberian flood volcanism. *Science*, *269*(5229), 1413–1416.
URL https://www.science.org/doi/abs/10.1126/science.269.5229.1413

Retallack, G. J., Sheldon, N. D., Carr, P. F., Fanning, M., Thompson, C. A., Williams, M. L., Jones, B. G., & Hutton, A. (2011). Multiple early triassic greenhouse crises impeded recovery from late permian mass extinction. *Palaeogeography, Palaeoclimatology, Palaeoecology*, *308*(1), 233–251. URL `https://www.sciencedirect.com/science/article/pii/S0031018210005924`

Romano, C., Goudemand, N., Vennemann, T. W., Ware, D., Schneebeli-Hermann, E., Hochuli, P. A., Brühwiler, T., Brinkmann, W., & Bucher, H. (2013). Climatic and biotic upheavals following the end-permian mass extinction. *Nature Geoscience*, *6*(1), 57–60.

Schneebeli-Hermann, E. (2020). Regime shifts in an early triassic subtropical ecosystem. *Frontiers in Earth Science*, *8*. URL `https://www.frontiersin.org/articles/10.3389/feart.2020.588696`

Stanley, S. M. (2016). Estimates of the magnitudes of major marine mass extinctions in earth history. *Proceedings of the National Academy of Sciences*, *113*(42), E6325–E6334. URL `https://www.pnas.org/doi/abs/10.1073/pnas.1613094113`

Sun, Y., Joachimski, M. M., Wignall, P. B., Yan, C., Chen, Y., Jiang, H., Wang, L., & Lai, X. (2012). Lethally hot temperatures during the early triassic greenhouse. *Science*, *338*(6105), 366–370. URL `https://www.science.org/doi/abs/10.1126/science.1224126`

Voigt, A., & Abbot, D. S. (2012). Sea-ice dynamics strongly promote snowball earth initiation and destabilize tropical sea-ice margins. *Climate of the Past*, *8*(6), 2079–2092. URL `https://cp.copernicus.org/articles/8/2079/2012/`

Wahid, A., Gelani, S., Ashraf, M., & Foolad, M. (2007). Heat tolerance in plants: An overview. *Environmental and Experimental Botany*, *61*(3), 199–223. URL `https://www.sciencedirect.com/science/article/pii/S0098847207000871`

Widmann, P., Bucher, H., Leu, M., Vennemann, T., Bagherpour, B., Schneebeli-Hermann, E., Goudemand, N., & Schaltegger, U. (2020). Dynamics of the largest carbon isotope excursion during the early triassic biotic recovery. *Frontiers in Earth Science*, *8*. URL `https://www.frontiersin.org/articles/10.3389/feart.2020.00196`

Zhu, F., & Rose, B. E. J. (2023). Multiple equilibria in a coupled climate–carbon model. *Journal of Climate*, *36*(2), 547 – 564. URL `https://journals.ametsoc.org/view/journals/clim/36/2/JCLI-D-21-0984.1.xml`